# Understanding Bias in Large-Scale Visual Datasets

**Boya Zeng**[*]
University of Pennsylvania

**Yida Yin**[*]
UC Berkeley

**Zhuang Liu**
Meta FAIR

[*]equal contribution

## Abstract

A recent study [40] has shown that large-scale visual datasets are very biased: they can be easily classified by modern neural networks. However, the concrete forms of bias among these datasets remain unclear. In this study, we propose a framework to identify the unique visual attributes distinguishing these datasets. Our approach applies various transformations to extract semantic, structural, boundary, color, and frequency information from datasets, and assess how much each type of information reflects their bias. We further decompose their semantic bias with object-level analysis, and leverage natural language methods to generate detailed, open-ended descriptions of each dataset's characteristics. Our work aims to help researchers understand the bias in existing large-scale pre-training datasets, and build more diverse and representative ones in the future. Our project page and code are available at boyazeng.github.io/understand_bias.

## 1 Introduction

Recently, Liu and He [40] revisited the "*Name That Dataset*" experiment introduced by Torralba and Efros [69] in 2011, which highlighted the built-in bias of visual datasets. It is a classification task where each dataset forms a class, and models are trained to predict the dataset origin of each image. The datasets back in 2011 were found to be classified quite accurately by SVMs [69]. Since then, significant efforts have been devoted to creating more diverse, large-scale, and comprehensive datasets [15, 37, 35, 60]. Surprisingly, a decade later, the largest and supposedly most diverse datasets (*e.g.*, YFCC [66], CC [11], DataComp [19]) can still be classified with remarkably high accuracy by modern neural networks [40].

Although we now know these large-scale datasets are very biased, a lingering question remains: what are the concrete forms of bias among them[1], that cause them to be easily classified? Understanding the bias among datasets is essential for addressing it, and for improving dataset diversity and coverage. Creating datasets that more comprehensively represent the real world is challenging yet crucial [64, 55, 31]—only then can we build truly general-purpose vision systems—systems capable of handling various scenarios out of the box, and performing reliably in real-world situations [30, 29, 24].

To this end, we develop a framework for understanding the concrete forms of bias among datasets. We isolate the semantic, structure, boundary, color, and frequency information through various transformations. For example, transforming an image into a semantic segmentation map preserves semantics while discarding most texture information. We then perform the dataset classification task on the transformed datasets, to quantify how each type of information reflects the dataset bias.

To pinpoint the semantic bias in datasets, we further conduct object-level and open-ended language analysis. Specifically, we leverage pre-trained recognition models to identify objects that characterize each dataset. In addition, using a Vision-Language Model (VLM), we generate image captions as

---

[1]Note that in this work we study bias *among* datasets. This concerns the coverage of concepts and content (*i.e.*, how representative the dataset is for the real world). This is related to, but different from, another notion of bias that concerns social and stereotypical bias, as well as algorithm fairness [8, 75, 6, 47].

38th Conference on Neural Information Processing Systems (NeurIPS 2024).

surrogate language representations of the images. We then apply topic models and Large Language Models (LLMs) to generate natural language descriptions for each dataset.

We apply our framework to three popular large-scale visual datasets: YFCC, CC, and DataComp, following [40]. Surprisingly, after transforming images into various semantic and structure representations (*e.g.*, object bounding boxes and contours), neural networks can almost always still accurately predict their dataset origin. This highlights the bias in semantics and object shapes. Our object-level queries further reveal a discrepancy in object diversity and distribution across the YCD datasets. Lastly, open-ended language analysis indicates that YFCC emphasizes outdoor and natural scenes with human interactions, while DataComp features digital graphics heavily.

Our framework operates on images only and does not require any human annotations, making it compatible with any image dataset. It can be applied in future dataset curation to assess data diversity, and guide the inclusion of data with various attributes. We hope this work can help researchers address dataset bias based on their needs, and develop more inclusive and diverse visual datasets.

## 2   Related Work

**Dataset classification**. The dataset classification problem was originally proposed by Torralba and Efros [69] in 2011. Tommasi *et al.* [68] later also studied this problem with linear classifiers using pre-trained CNN features. In contrast to prior work focusing on labeled smaller-scale datasets with shared object classes, Liu and He [40] recently revisited the dataset classification problem with large-scale, diverse, and presumably less-biased datasets. They demonstrated that modern neural networks are still excellent at capturing bias among these datasets. Our work identifies the exact forms of the bias among datasets beyond dataset classification results.

**Social bias and fairness**. Extensive literature has highlighted that various visual datasets underrepresent certain demographic groups [70, 75, 51], contain various gender stereotypes [81, 28, 45], or ignore some geographical regions [62, 75, 71]. These social fairness issues in datasets result in the deployment of flawed models [8, 72, 6] that may produce biased predictions or struggle to generalize well across different domains. Instead of focusing on social fairness in each dataset, we study the representativeness (*i.e.*, coverage of real-world concepts and objects) and understand its differences among datasets. Note that Meister *et al.* [45] also use transformations to isolate different types of information, with a specific focus on gender bias in datasets.

**Bias detectors and debiasing tools**. There are approaches that can locate the imbalance of object representation within datasets [23, 71]. Dataset rebalancing methods [8, 75] seek to correct these representation imbalances across protected attributes. Algorithmic intervention and regularization, such as adversarial training [78, 79, 44] and domain-independent training [73], can counteract the propagation of bias and stereotypes in downstream modeling. Note that these prior methods often require ground-truth annotations to identify or mitigate potential bias, while our framework can analyze unlabeled pre-training datasets and provide insights beyond object distribution imbalances.

## 3   Isolating Bias with Transformations

Although modern neural networks can achieve excellent accuracy in the dataset classification problem [40], what bias is captured by neural networks remains unclear. To better understand this, we selectively preserve or suppress specific types of information using various transformations. We then train a new model on the transformed datasets to perform the dataset classification task. As a result, its dataset classification performance indicates the level of bias in the extracted information. For example, transforming an image into a depth map captures the spatial geometry while discarding texture, allowing us to assess bias present solely in spatial information.

### 3.1   Datasets and Settings

Based on Liu and He [40], we take YFCC100M [66], CC12M [11], and DataComp-1B [19] (collectively referred to as "YCD") and study their bias in this work. Figure 1 shows example images.

Our training setup is also adapted from [40]. Specifically, we randomly sample 1M and 10K images from each dataset as training and validation sets, respectively. We employ the same ConvNeXt-Tiny

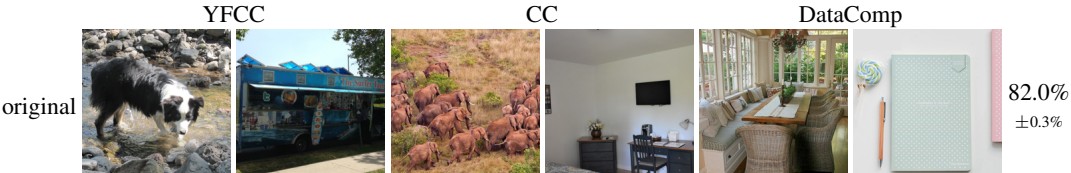

| | YFCC | CC | DataComp | |
|---|---|---|---|---|
| original | | | | 82.0%
±0.3% |

Figure 1: **Original images**. We sample two images from each of YFCC [66], CC [11], and DataComp [19]. Dataset classification on the original images has a reference accuracy of 82.0%.

image classification model [41] and train it for 30 epochs to classify the combined dataset with 3M samples. Note that the original work [40] achieves 84.7% accuracy on the validation set, while we achieve 82.0% accuracy due to shorter training (roughly 23% of original length). We refer to this 82.0% as the **reference accuracy** in this paper. For all image transformations, we use the same ConvNeXt-Tiny model and almost identical recipes (details in Appendix A.1). We repeat data sampling and experiments three times, reporting mean validation accuracy and standard deviation.

## 3.2 Semantics

To start, we seek to understand how semantically biased the datasets are. Specifically, we extract semantic components from the images with fine-grained, coarse-grained, or no spatial detail. Also, we apply a variational autoencoder, potentially reducing low-level signatures (*e.g.*, color quantization, JPEG compression). Figure 2 shows the transformations and their dataset classification accuracies.

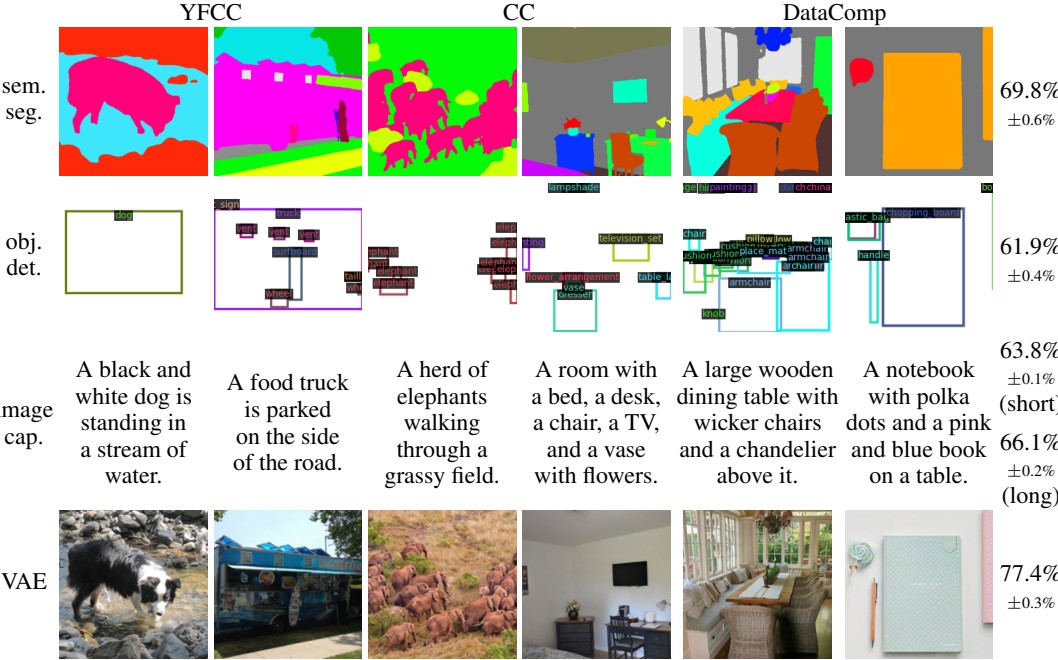

Figure 2: **Transformations preserving semantic information** (semantic segmentation, object detection, and caption) **and potentially reducing low-level signatures** (VAE) result in high dataset classification accuracy. This suggests that semantic discrepancy is an important form of dataset bias.

**Semantic segmentation** offers fine-grained semantic annotation with rich object information by assigning a class label to each pixel. We take a semantic segmentation model ViT-Adapter-Large [12] (with BEiT-v2 [53] backbone) trained on ADE20K [83] with 150 semantic classes (*e.g.*, wall, building, sky, *etc.*) to generate a semantic segmentation mask for each image. This mask is represented as an RGB image using a color palette for different classes, as shown in Figure 2. The model trained on this color-coded mask achieves 69.8% accuracy, well above the chance level.

**Object detection** provides coarse spatial annotations for objects through rectangle bounding boxes. We use ViTDet-Huge [36] trained on LVIS [25] with 1203 object categories to derive bounding boxes from each image. We keep object class names on the bounding boxes to account for semantic meaning. This representation reaches 61.9% dataset classification accuracy, below semantic segmentation.

**Image captioning** discards all visual information and produces semantic representations through natural language. Captions are less affected by pixel variations and spatial cues in images. Using LLaVA 1.5 [39, 38], we generate a single-sentence caption and a long-paragraph caption for each image. Short captions are shown in Figure 2, and long captions are in Appendix D. We finetune a sentence embedding model Sentence T5-base [50] to perform dataset classification on these captions. This results in 63.8% accuracy on short captions and 66.1% on long ones, both nearing the accuracy for semantic segmentation. The richer details in longer captions enhance performance.

**Variation autoencoder (VAE)** [33] encodes each image into a latent vector and then reconstructs the original image from it. The datasets may use different JPEG compressions or image resolutions, which could be exploited as shortcuts by dataset classification models. However, VAE's low-dimensional latent space may encode semantic information and suppress such low-level signatures. Reconstructing the images with a pre-trained VAE from Stable Diffusion [58] only slightly decreases the accuracy from 82.0% to 77.4%, suggesting that low-level bias may have a smaller impact than semantic bias.

Semantic segmentation, object detection, and image captioning extract semantic information with decreasing levels of spatial information. On the other hand, VAE could potentially reduce low-level signatures while preserving the semantics. The high accuracies of dataset classification models indicate that *semantic bias is an important component of dataset bias in YCD*.

### 3.3 Structures

Next, we analyze the dataset bias rooted in object shape and spatial geometry rather than object semantics. To capture such structural visual bias, we use the Canny edge detector and the Segment Anything Model (SAM) [34] to outline object contours, and the Depth-Anything-V2 model [76] to measure pixel-level depth. While contour delineates fine-grained object shape, and depth estimation offers relative object positions, both lack the rich object semantic details present in semantic segmentation masks and bounding boxes (*e.g.*, object class). Figure 3 visualizes the transformations.

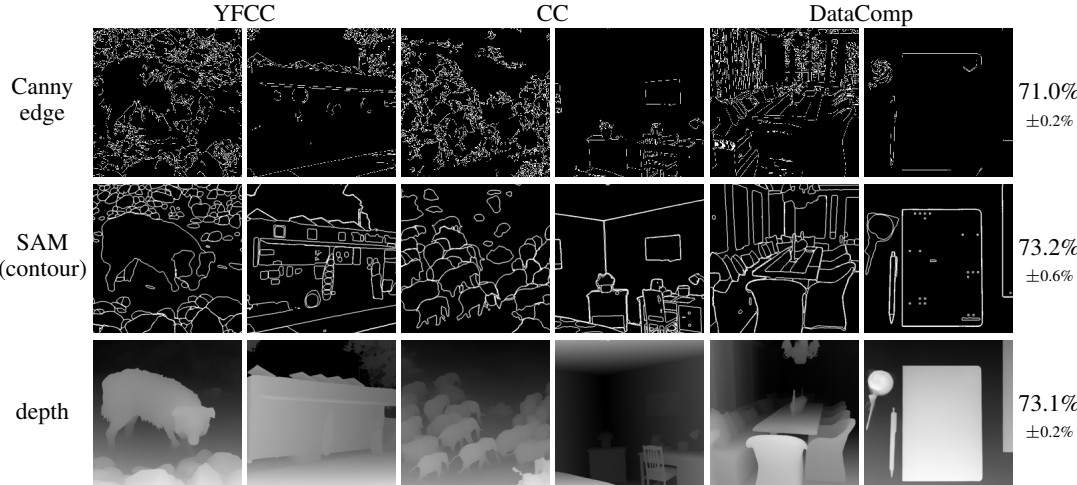

Figure 3: **Transformations outlining object shapes and estimating pixel depth**. Dataset classification achieves even higher accuracies on object contours and depth images than on semantic information, indicating that object shapes and spatial geometry vary significantly across YCD.

**Canny edge detector** [10] is a classical algorithm that outlines rough object boundaries by capturing sharp intensity changes. It removes noise with a Gaussian filter, calculates intensity gradients, and applies non-maximum suppression to form edges, represented as a binary mask. This results in 71.0% classification accuracy, 11% below the reference accuracy (82.0%).

**Segment Anything Model (SAM)** [34] can provide high-quality object segmentation masks. We could then use them to delineate cleaner and more accurate shapes of objects that are minimally affected by local pixel variations, compared to the Canny edge detector. Specifically, we use SAM with the ViT-Large backbone to generate class-agnostic segmentation masks and identify boundaries by finding pixels whose surrounding pixels are not all from the same object. The classification accuracy on SAM contours (73.2%) is slightly higher than that on Canny edge (71.0%).

**Depth** estimation captures the scene's spatial geometry, offering fine-grained spatial context and relative object positioning. Like contours, it excludes explicit semantic information about objects. The Depth-Anything-V2 (ViT-L) model [76] generates pixel-level depth estimation, encoded as a normalized grayscale image. The resulting 73.1% accuracy is comparable to that of SAM contours.

The dataset classification accuracies on object contours and depth are even higher than the ones on semantics. This shows that *object shape and spatial geometry variations are significant among YCD.*

### 3.4 Spatial Permutations

To further understand the level of bias captured in spatial information as opposed to semantics, we keep the RGB values of all pixels in each image unchanged but shuffle the pixel positions to disrupt spatial coherence. We shuffle each image on the pixel level and the patch level, following a fixed order and a random order for all images. Figure 4 shows the images shuffled in a random order.

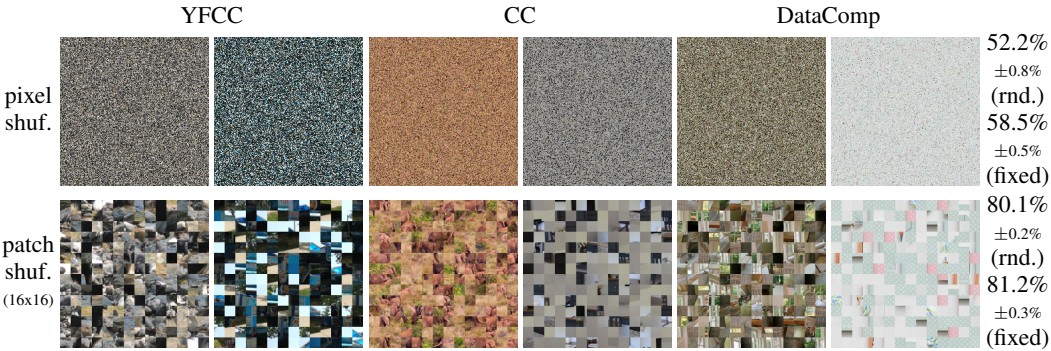

|  | YFCC | CC | DataComp |  |
|---|---|---|---|---|
| pixel shuf. | | | | 52.2% ±0.8% (rnd.) 58.5% ±0.5% (fixed) |
| patch shuf. (16x16) | | | | 80.1% ±0.2% (rnd.) 81.2% ±0.3% (fixed) |

Figure 4: **Transformations breaking spatial structure**. Pixel shuffling drastically decreases dataset classification accuracy, but patch shuffling has minimal impact. This demonstrates that local structure is important and sufficient for models to learn the patterns of each dataset.

**Pixel shuffling** obfuscates the image classifier with a permutation of the pixels and forces it to find patterns from the color distribution of pixels in each image. As expected, this significantly decreases the classification accuracy to 52.2% for the random order and 58.5% for the fixed order.

**Patch shuffling** first divides each image into smaller patches and then rearranges the order of the patches. Consequently, it preserves more local spatial information. Here we vary the patch size and show the results in Figure 5. The accuracies of the fixed order and the random order shuffling are almost identical when the patch size is larger than 1. Surprisingly, with a patch size of 16, we almost reach the 82% reference accuracy.

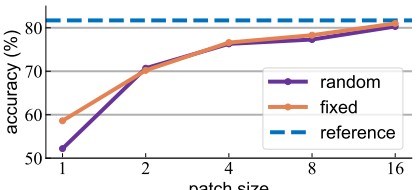

Figure 5: **Effect of patch sizes**. Dataset classification accuracy approaches the reference one with larger patch sizes.

The significant performance drop with pixel shuffling shows *completely destructing the local structure in YCD can reduce its dataset bias to a large extent*. However, the minimal accuracy decrease after shuffling patches of size 16 indicates *patch-level local structures in spatial information is sufficient for identifying visual signatures of the YCD datasets.*

### 3.5 RGB

The high classification accuracy after pixel shuffling implies a discrepancy in pixel color distributions among YCD. To further assess this difference in color statistics among datasets, we transform each image into its average value for each color channel. Figure 6 shows the resulting images.

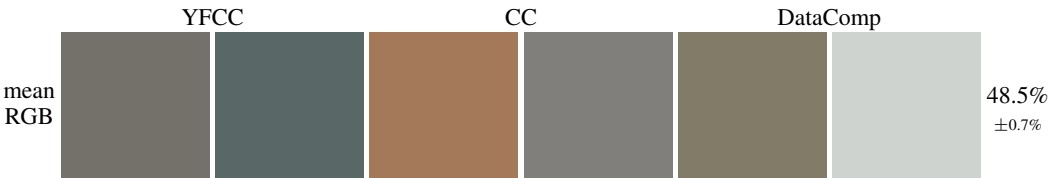

Figure 6: **Averaging each color channel**. Even when the values of each channel in images are averaged, the model can still achieve non-trivial dataset classification performance.

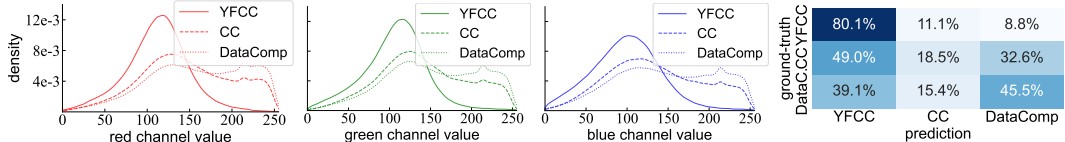

Figure 7: **Distribution of mean RGB values and confusion matrix**. YFCC's RGB values are overall smaller, while CC's and DataComp's are very similar. This is also reflected in the confusion matrix of dataset classification on mean RGB images, where YFCC can be classified very easily (indicated by the dark blue box on the top left), while there is high confusion between CC and DataComp.

**Mean RGB**. We compute the mean RGB value for each image. This abstracts the pixel details into a constant RGB color map and forces the image classifier to only use color statistics. The model's accuracy on mean RGB images is 48.5%, about 15% higher than the chance-level accuracy of 33.3%.

In Figure 7, we visualize the distribution of mean RGB values for YCD. There is only a moderate difference in mean RGB distribution between CC and DataComp. However, *YFCC is much darker than CC and DataComp.* This is further suggested by the confusion matrix of the dataset classification model trained on mean RGB images shown in Figure 7, where the model classifies YFCC accurately but shows more confusion when distinguishing between CC and DataComp.

### 3.6 Frequency

Neural networks tend to exploit texture patterns even when the recognition task is inherently about the semantics [32, 21, 20]. If we decompose visual signals by frequencies, high-frequency bands typically capture these texture patterns and sharp transitions, whereas low-frequency components represent general structure and smooth variations. To explore how different frequency components contribute to dataset bias, we apply high-pass and low-pass filters to the original images.

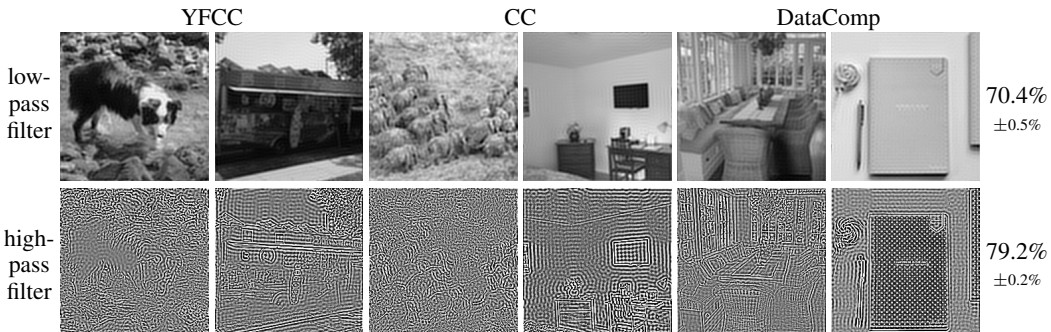

Figure 8: **Transformations filtering high-frequency and low-frequency components** retain close-to-reference accuracy. This indicates that dataset bias exists in different frequencies. The high-pass filtered images are equalized for better visualization.

**High-pass filter and low-pass filter**. To filter signals based on frequencies, we first perform a 2D Fast Fourier Transform on each grayscaled image to obtain its representation in the frequency domain. We then apply an ideal filter [22] with a hard threshold radius of 40 in the frequency domain, so as to only keep either high (*i.e.*, high-pass filter) or low (*i.e.*, low-pass filter) frequencies. The filtered results are finally inversely transformed to the original grayscale domain, visualized in Figure 8. Additional results and visualizations are in Appendix B.6.

The model trained on images with high frequencies kept has an accuracy of 79.2%. This is slightly better than the one trained on images with low frequencies kept, which has an accuracy of 70.4%. Both accuracies are close to the reference accuracy of 82.0%.

The high accuracy of models trained on either frequency component indicates that *visual bias in the YCD datasets exists in both low-frequency and high-frequency components.*

### 3.7 Synthetic Images

Synthetic images hold significant potential in augmenting data for various vision tasks [27, 3, 67, 1, 5]. Diffusion models [56, 46, 58] can generate synthetic images. However, if dataset bias can be inherited from a diffusion model's training images to its generated images, bias may persist and even amplify in downstream tasks that use generated images for training. To assess the bias in synthetic images, we run dataset classification on images generated from diffusion models, illustrated in Figure 9.

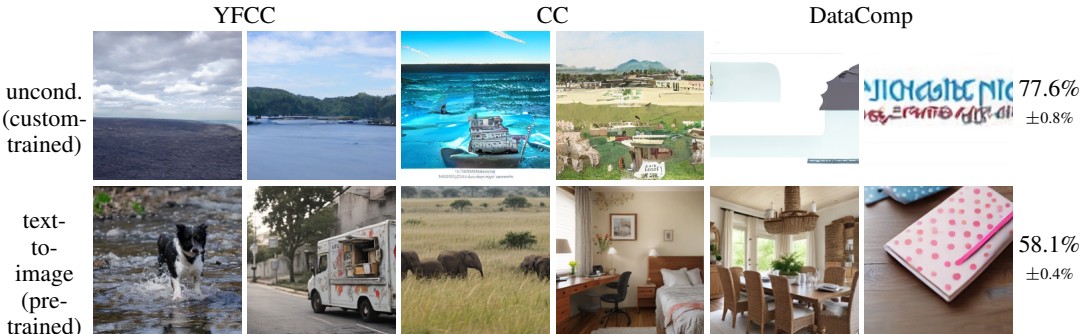

Figure 9: **Synthetic images from unconditional and text-to-image generation models**. Unconditionally generated images can be classified with near-reference accuracy. Images from text-to-image diffusion models using short captions have reasonable dataset classification accuracy.

**Unconditional generation**. We train an unconditional Diffusion Transformer (DiT) [52] on each individual dataset in YCD. The models learn to generate synthetic images from random noise. Dataset classification is performed on the combination of synthetic data generated from each model, resulting in a very high accuracy of 77.6%, nearly matching the reference accuracy of 82.0%.

**Text-to-Image** generation on image captions potentially preserves the semantic bias in the original images. We generate synthetic images from the SDXL-Turbo [59] diffusion model, conditioned on short captions produced by LLaVA (Section 3.2). By converting the original images into caption text and then back to the visual domain, we only retain the semantics captured in captions. Note that, unlike the unconditional generation experiment above, here we do not train our own text-to-image model for each dataset; instead, we use the same pre-trained model for all datasets. We reach 58.1% accuracy, falling slightly short of 63.8% when directly classifying short captions.

The high classification accuracy from unconditionally generated images shows that *synthetic images sampled from a diffusion model can inherit the bias in the model's training images.* The ability to classify synthetic images generated by pre-trained text-conditional generation model further confirms that *semantic discrepancy is a major contributor to dataset bias.*

*Summary*. We analyzed the impact of various transformations on dataset classification, identifying semantics and structures as important contributors to dataset bias. Patch-level local structure information is sufficient to classify YCD, with datasets differing even in color statistics. Bias spans across frequency components, particularly in high-frequency bands. Finally, we showed that bias can be inherited in synthetic images of diffusion models. More results are in Appendix B.

## 4 Explaining Semantic Bias among Datasets

In the preceding section, we explore transformations to extract various types of image information, each exhibiting varying levels of bias. Among them, semantic bias heavily contributes to the high accuracy in the dataset classification task [40]. In this section, we identify specific interpretable semantic patterns within each dataset through object-level and language-based analysis.

## 4.1 Object-level Queries

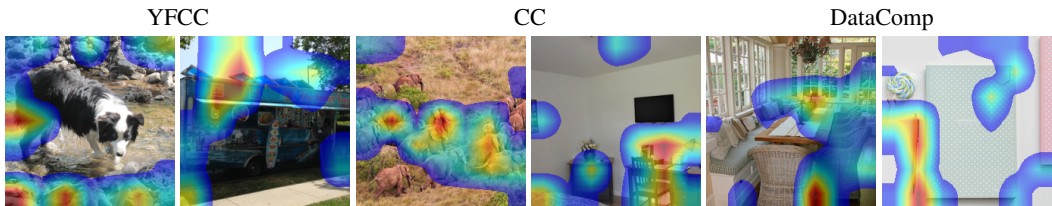

YFCC          CC          DataComp

Figure 10: **Grad-CAM heatmap** [82, 61] on the dataset classification model trained on original datasets. The model focuses on specific objects to determine the dataset origin of each image.

Grad-CAM [82, 61] highlights key regions in an input image that explain neural network predictions. In Figure 10, applying Grad-CAM to the reference model (Section 3.1) shows that the model focuses on semantically meaningful objects: elephant herd in the third image, table and chair in the fourth image, and pen in the sixth image. This suggests that the model might have leveraged the object-level information to recognize the dataset identity of each image. To better understand this, we apply models pre-trained on other vision datasets (ImageNet-1K [15], LVIS [25], and ADE20K [83]) to provide object annotations for each YCD image, and analyze their object-level bias. As a result, the analysis below is in the context of 3 sets of object classes, defined by these 3 datasets.

**Imbalanced object distribution**. Imbalance in object distribution is a common form of semantic bias. For each object class, we calculate the number of images in each dataset in YCD that contain the object class and their percentage share relative to all images with that object class. Note for LVIS and ADE20K models' output, we count each object class only once per image, even if multiple instances or pixels of the same object class are present. Figure 11 shows the top 8 object classes with the highest percentage of images from YFCC, CC, or DataComp. The dominance of a certain dataset within these classes highlights a considerable imbalance in object-level distribution across datasets.

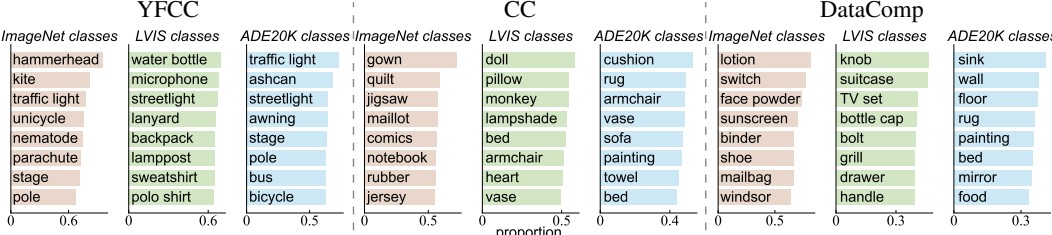

Figure 11: **Object classes with the highest proportions of YFCC, CC, or DataComp images**. Less-frequent classes are not shown. Most classes consist predominantly of images from one dataset.

Figure 11 also shows that YFCC constitutes much higher proportions in its top object classes than CC and DataComp in their respective classes (note the different x-axis scales in each subplot). To further see this, we visualize the distribution of unique object class counts per image in Figure 12. The higher variety of objects in YFCC images shows a notable gap in object diversity among YCD.

**Interpretable dataset classification with objects**. The coefficients of a logistic regression model form a natural importance ranking of input features when the features are binary. To leverage this, we represent each image with a binary vector, where each element indicates the presence of a specific object class from a set of objects (*i.e.*, ImageNet, LVIS, or ADE20K). We train a logistic regression model to predict the dataset origin of the images based on their binary vector representations. This simple model achieves validation accuracies of 52.0% with ImageNet objects, 52.4% with LVIS objects, and 52.4% with ADE20K objects.

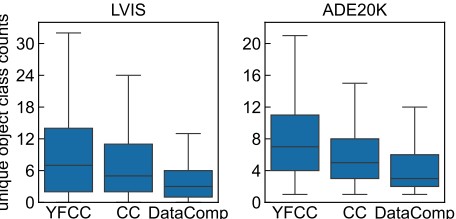

Figure 12: **Unique object classes per image**. On average, YFCC contains the highest number of unique objects in each image, followed by CC, while DataComp exhibits the lowest.

Figure 13 shows the top objects based on logistic regression coefficients. It highlights outdoor infrastructures (*e.g.*, traffic light, clock tower, telephone pole, and building) in YFCC and household

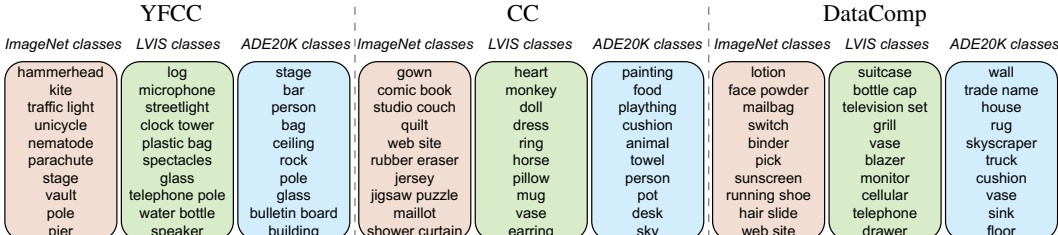

| YFCC | | | CC | | | DataComp | | |
|---|---|---|---|---|---|---|---|---|
| *ImageNet classes* | *LVIS classes* | *ADE20K classes* | *ImageNet classes* | *LVIS classes* | *ADE20K classes* | *ImageNet classes* | *LVIS classes* | *ADE20K classes* |
| hammerhead | log | stage | gown | heart | painting | lotion | suitcase | wall |
| kite | microphone | bar | comic book | monkey | food | face powder | bottle cap | trade name |
| traffic light | streetlight | person | studio couch | doll | plaything | mailbag | television set | house |
| unicycle | clock tower | bag | quilt | dress | cushion | switch | grill | rug |
| nematode | plastic bag | ceiling | web site | ring | animal | binder | vase | skyscraper |
| parachute | spectacles | rock | rubber eraser | horse | towel | pick | blazer | truck |
| stage | glass | pole | jersey | pillow | person | sunscreen | monitor | cushion |
| vault | telephone pole | glass | jigsaw puzzle | mug | pot | running shoe | cellular | vase |
| pole | water bottle | bulletin board | maillot | vase | desk | hair slide | telephone | sink |
| pier | speaker | building | shower curtain | earring | sky | web site | drawer | floor |

Figure 13: **Object class ranking from logistic regression coefficients**. The regression classifies images based on object presence. YFCC has more top objects related to outdoor scenes, while CC and DataComp focus on household items and products. Classes with low frequencies are not shown.

items, products, and digital graphics (*e.g.*, doll, ring, vase, blazer, and website) in CC and DataComp. These rankings partially overlap with the list of objects that are much more prevalent in one dataset than the others (Figure 11). However, the object rankings also identify objects that are more balanced across datasets, since logistic regression receives more weight updates based on more common objects during training. Thus, it provides a complementary angle on object-level bias among the datasets.

## 4.2 Open-ended Language Analysis

The word clouds [48] in Figure 14 offer a visual representation of the most prevalent phrases in the long paragraph captions of each dataset from Section 3.2. We observe several frequent phrases featuring human subjects (*e.g.*, people, group, wearing) in YFCC, elements of indoor scenes (*e.g.*, room, and dining table) in CC, and a focus on white background in DataComp. In this section, we will use the long captions as a proxy to dive deeper into the semantic themes in each dataset.

**Unsupervised topic discovery**. We treat each caption as a document and apply the Latent Dirichlet Allocation (LDA) [7] for topic discovery to each dataset in YCD (with the number of topics set to 5). Figure 15 presents the top 5 words for each topic. Notably, in YFCC, three topics (first, second, and fifth) feature words associated with outdoor scenes; in CC and DataComp, their topics cover "logo" and "design," suggesting the presence of digital graphics.

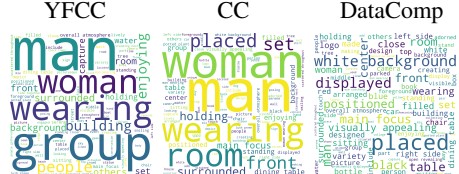

Figure 14: **Word clouds** [48] on the 100 most frequent phrases in each dataset. Phrase size corresponds to its frequency.

| YFCC | CC | DataComp |
|---|---|---|
| [building, scene, street, car, sign] | [room, table, chairs, chair, dining] | [logo, background, design, book, colors] |
| [scene, field, game, person, dog] | [design, background, colors, logo, display] | [scene, table, room, building, atmosphere] |
| [table, room, dining, scene, items] | [woman, man, shirt, scene, dress] | [car, scene, truck, background, kitchen] |
| [people, man, woman, scene, group] | [scene, people, water, group, atmosphere] | [background, table, design, box, bottle] |
| [scene, water, sky, trees, tree] | [scene, building, car, person, dog] | [man, woman, scene, shirt, person] |

Figure 15: **LDA-extracted topics** for each generated caption set. Each row lists the top 5 words for a topic. YFCC focuses on outdoor scenes, while CC and DataComp contain more digital graphics.

**Large Language Model (LLM) summarization**. To assess whether dataset bias in captions can be identified by LLMs with limited examples, we provide GPT-4o with 120 captions per dataset and ask it to infer the dataset origin of new captions. Inference is performed on one validation caption at a time until accuracy stabilizes at 52.9% on 480 captions. Further details are in Appendix A.2.

This powerful ability of LLMs allows them to provide open-ended and detailed natural language explanations. Specifically, we procedurally prompt GPT-4o to extract dataset-specific characteristics from caption datasets and refine its answers into 5 bullet points, shown in Figure 16. In summary, YFCC is characterized by abundant outdoor, natural, and human-related scenes, while DataComp concentrates on static objects and digital graphics with clean backgrounds and minimal human presence. In contrast, CC blends elements of both YFCC's dynamic scenes and DataComp's static imagery. Appendix A.2 provides the entire prompt structure and the complete LLM summarization output. We further verify the validity of the semantic features from LDA and LLM in Appendix C.

| YFCC | CC | DataComp |
|------|-----|----------|

**1. Group Dynamics and Activities**
This distribution frequently showcases groups of people engaged in activities such as playing music, attending events, or participating in sports, emphasizing social interactions and communal settings.

**2. Urban and Social Settings**
Captions often describe dynamic environments filled with people and activity in urban or public settings, such as busy city streets, transportation hubs, and social events.

**3. Serene Natural Settings**
Many images feature serene outdoor environments, including natural landscapes, gardens, and bodies of water, highlighting a calm and peaceful atmosphere.

**4. Detailed Environmental Context**
...

**5. Emotions and Interactions**
...

**1. Organized Indoor and Outdoor Scenes**
Captions depict well-structured environments, including cozy bedrooms, dining areas, cityscapes, and architectural landmarks, emphasizing the arrangement and detail.

**2. Human Interactions and Social Events**
Emphasis on social and formal gatherings like weddings, concerts, and ceremonies, highlighting attire, decor, and the lively atmosphere.

**3. Vivid and Dynamic Elements**
Descriptions focus on colorful and lively scenes, with vibrant attire, festive settings, and active engagements, emphasizing visual appeal and movement.

**4. Detailed Objects and Clothing**
...

**5. Creative and Artistic Themes**
...

**1. Object-Focused Descriptions**
Captions prominently feature specific objects or products (e.g., coffee mugs, toys, cars), often isolated against minimalistic backgrounds to highlight their characteristics.

**2. Vibrant and Playful Visuals**
Scenes frequently include vibrant, colorful, and playful elements, focusing on visually appealing and lively imagery that captures attention.

**3. Close-Up and Detailed Views**
Descriptions often emphasize close-up shots, highlighting the intricate details, textures, and designs of objects, with a focus on aesthetic and functional attributes.

**4. Serene and Artistic Compositions**
...

**5. Simplistic and Isolated Backgrounds**
...

Figure 16: **LLM summarization of dataset features**. The bullet points highlight outdoor, natural, and human scenes in YFCC and static objects and synthetic images in DataComp. CC contains both YFCC's dynamic scenes and DataComp's static images.

*Summary*. We leveraged closed-set object-level queries and open-ended language analysis to interpret the semantic bias among datasets. The object-based analysis identified objects indicative of each dataset within a predefined object set. On the other hand, natural language methods are able to provide open-ended explanations for the characteristics of each dataset with rich details.

# 5 Discussion

YCD have different sources: YFCC is selected with minimal filtering from user-uploaded images on Flickr.com, while CC and DataComp filter web-sourced images with high quality in caption, image, or their alignment. We build on the interpretable semantic bias in Section 4 to discuss the dataset curation procedures, and provide potential suggestions.

*Filtering based on a reference dataset or model may inherit its bias*. DataComp has the fewest unique objects per image (Figure 12). This is possibly because DataComp filters for images with visual content close to ImageNet data in the embedding space [19]. Therefore, the remaining images tend to be object-centric [4]. It also filters for images that align well with its captions in CLIP [54] embedding space, therefore favoring certain types of images, *e.g.*, images containing text. To mitigate this, we may use datasets or models that contain less bias themselves for filtering.

*The source website's image collection mechanism can introduce bias*. We note that YFCC is heavily skewed towards outdoor scenes and human interactions (Section 4.2). This bias likely stems from its reliance on a single data source, Flickr.com, where user-uploaded content often focuses on personal photos, landscapes, and social interactions.

*Web images naturally contain more digital graphics*. Since CC and DataComp images are from Internet webpages, professionally created content like advertisements, infographics, and digital media are prioritized. Dataset users should evaluate if this composition aligns with the downstream goals.

# 6 Conclusion

We proposed a framework to study the bias in large-scale visual datasets and used it to analyze three representative datasets. By classifying transformed images' dataset origin, we identified structures and semantics as key factors in dataset bias. We further investigated specific forms of semantic bias among datasets through fixed object queries, highlighting distinctive concepts characterizing each dataset. Lastly, we extracted key topics and natural language summaries for each dataset. We hope this framework and these findings can encourage further exploration of dataset bias and help improve diversity and representation in future datasets.

**Acknowledgements**. We would like to thank Kaiming He, Olga Russakovsky, Mingjie Sun, Kirill Vishniakov, and Zekai Wang for helpful discussions and feedback.

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

# Appendix

## Table of Contents

# A   Implementation Details

## A.1   Dataset Classification with Transformations

**Dataset classification training**. Following Liu and He [40], for image-text datasets CC and Data-Comp, we only use their images. For each dataset in the YCD combination, we uniformly sample 1M / 10K to form the train / val sets. To speed up image loading, the shorter side of each image is resized to 500 pixels if the original shorter side is larger than this, with the aspect ratio preserved. We observe that this resizing has minimal effect on model performance. The dataset classification model is trained for 30 epochs, 23% of the original training length in Liu and He [40]. Table 1 details our default training recipe for dataset classification in Section 3.

| config | value |
|---|---|
| optimizer | AdamW [42] |
| learning rate | 1e-3 |
| weight decay | 0.3 |
| optimizer momentum | $\beta_1, \beta_2 = 0.9, 0.95$ |
| batch size | 4096 |
| learning rate schedule | cosine decay |
| warmup epochs | 2 |
| training epochs | 30 |
| augmentation | `RandomResizedCrop` [65] & `RandAug` (9, 0.5) [13] |
| label smoothing | 0.1 |
| mixup [80] | 0.8 |
| cutmix [77] | 1.0 |

Table 1: **Training recipe** for dataset classification.

During training, the model receives randomly augmented 224×224 image crops as input. At inference time, each image is first resized so that its shorter side has 256 pixels, with the aspect ratio maintained. A 224×224 center crop is then extracted and used as the model's input.

**Data processing**. By default, we use `RandomResizedCrop` and `RandAug` as data augmentations in training. Nevertheless, the data augmentations may need adjustments for certain image transformations. Specifically, we have the following two design choices: (1) We apply most transformations *before* data augmentations to avoid the time cost of transforming every augmented image. However, for pixel / patch shuffling and low-pass / high-pass filter, we apply these transformations *after* data augmentations. This is because patch sizes in patch shuffling often cannot evenly divide the original

| transformation | transformation before data augmentation | use `RandAug` |
|---|---|---|
| original | N/A | ✓ |
| semantic segmentation | ✓ | ✗ |
| object detection | ✓ | ✓ |
| VAE | ✓ | ✓ |
| Canny edge detector | ✓ | ✗ |
| SAM contour | ✓ | ✗ |
| depth | ✓ | ✗ |
| pixel shuffling | ✗ | ✓ |
| patch shuffling | ✗ | ✓ |
| RGB mean | ✓ | ✓ |
| low-pass filter | ✗ | ✗ |
| high-pass filter | ✗ | ✗ |
| unconditional generation | ✓ | ✓ |
| text-conditional generation | ✓ | ✓ |
| SIFT | ✓ | ✗ |
| HOG | ✓ | ✗ |

Table 2: **Data augmentation** details for classifying transformed images.

image dimensions. Moreover, shuffling with a fixed patch size on original images of varying dimensions before augmentations leads to inconsistent patch sizes in the final $224 \times 224$ augmented images. For high-pass / low-pass filter, image augmentations alter the frequency of visual signals, so applying these filters after augmentations ensures the intended frequency range is preserved. (2) `RandAug` is not used for images transformed into binary or grayscale formats (*e.g.*, Canny edge detector and low-pass filter) and those represented by color palettes (*e.g.*, semantic segmentation). This is because brightness and contrast adjustments in `RandAug` are designed for standard RGB images. Table 2 shows whether we apply transformation *before* data augmentation and whether we use `RandAug`.

**Other details**. In Section 3.2, we use LLaVA 1.5 [39, 38] with 4-bit quantization [16] for image captioning. The quantization minimally degrades performance. To perform the dataset classification on the generated captions, we finetune the Sentence T5-base [50] model using a batch size of 128. We search over learning rates {1e-3, 1e-4, 1e-5} and numbers of training epochs {1, 2, 4, 6}. The number of warmup iterations is set to 6% of the total training iterations. For VAE, we use the *KL-regularized VAE* [58] with a downsampling factor of 4, which encodes an RGB image of shape $256 \times 256 \times 3$ into a latent vector of size $64 \times 64 \times 3$. In Section 3.7, we train a DiT-B/2 [52] model on each of YCD datasets for 275K iterations, using a batch size of 1024 and a constant learning rate of 1e-4.

## A.2 LLM-based Analysis

In Section 4.2, we leverage LLMs to perform dataset classification with in-context learning on LLaVA-generated captions of YCD datasets and summarize the characteristics of each dataset. GPT-4o is the default LLM in our analysis. We accessed it via ChatGPT Temporary Chat in May 2024.

> Here are some samples from three different distributions of captions.
>
> {Caption 1} is from distribution {1, 2, or 3}
> {Caption 2} is from distribution {1, 2, or 3}
> ...
> {Caption 360} is from distribution {1, 2, or 3}
>
> Determine which of the three distributions is this caption sampled from:
> {validation caption}

Figure 17: **In-context learning prompt**. We provide an LLM with 360 in-context demonstrations, comprising 120 captions sampled from each of the YCD datasets. The model is then prompted it to predict the dataset origin of a hold-out caption. To prevent the LLM from using any prior knowledge of the YCD datasets, each dataset is anonymized as "distribution 1 / 2 / 3".

**In-context learning**. Figure 17 shows our in-context learning prompt. We create 360 in-context examples by sampling 120 long captions from each of the YCD datasets. Based on these in-context examples, the LLM needs to determine the source dataset of a hold-out caption. To avoid the LLM using any prior knowledge about the YCD datasets, we randomly assign an index from {1, 2, 3} to represent each dataset. We repeat the entire process with different in-context examples and evaluate the classification accuracy averaged across different hold-out samples. As shown in Figure 18, the accuracy stabilizes as the number of samples from each dataset reaches 160.

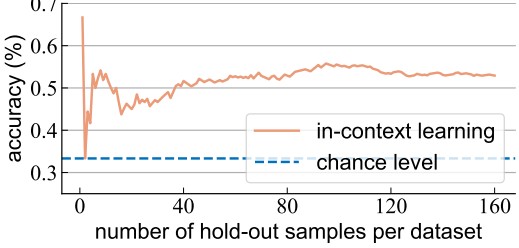

Figure 18: **The accuracy of in-context learning** converges at 160 samples per dataset.

**Summarization for dataset characteristics**. We employ a two-step process for an LLM to summarize the unique characteristics of each dataset. Figure 19 illustrates our two-step procedure and

the prompts. First, we provide the LLM with a combination of 360 randomly sampled captions (120 per dataset) and ask it to identify 2 distinctive patterns for each dataset. We repeat this process 10 times with different samples, generating 20 patterns for each dataset. Second, the LLM condenses the 20 patterns into 5 bullet points that characterize each dataset. Figure 20 shows the full LLM summarization of each dataset's characteristics.

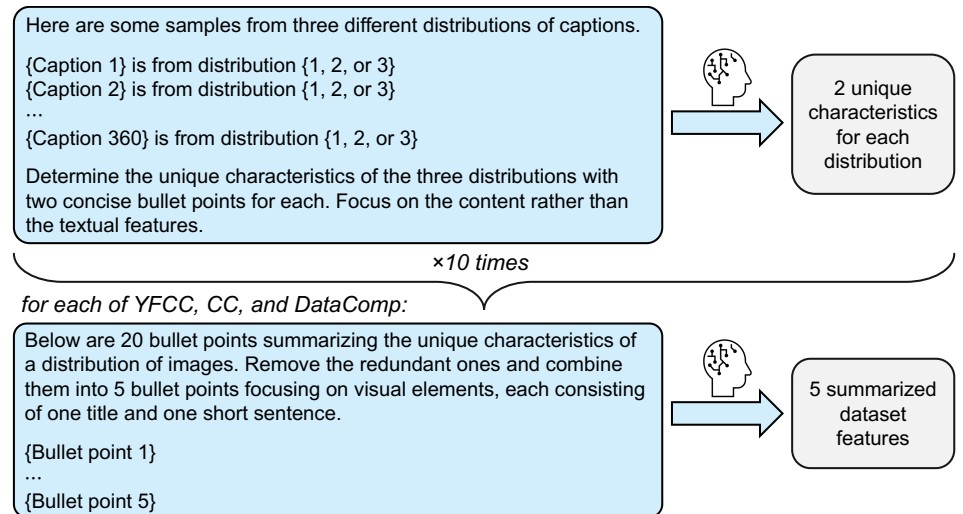

Figure 19: **Prompting procedure for LLM summarization**. The LLM summarizes the dataset characteristics over 10 iterations, with 360 randomly sampled captions per iteration. These characteristics are then aggregated, and the LLM consolidates them into five bullet points for each dataset.

| YFCC | CC | DataComp |
|---|---|---|
| **1. Group Dynamics and Activities** This distribution frequently showcases groups of people engaged in activities such as playing music, attending events, or participating in sports, emphasizing social interactions and communal settings. | **1. Organized Indoor and Outdoor Scenes** Captions depict well-structured environments, including cozy bedrooms, dining areas, cityscapes, and architectural landmarks, emphasizing the arrangement and detail. | **1. Object-Focused Descriptions** Captions prominently feature specific objects or products (e.g., coffee mugs, toys, cars), often isolated against minimalistic backgrounds to highlight their characteristics. |
| **2. Urban and Social Settings** Captions often describe dynamic environments filled with people and activity in urban or public settings, such as busy city streets, transportation hubs, and social events. | **2. Human Interactions and Social Events** Emphasis on social and formal gatherings like weddings, concerts, and ceremonies, highlighting attire, decor, and the lively atmosphere. | **2. Vibrant and Playful Visuals** Scenes frequently include vibrant, colorful, and playful elements, focusing on visually appealing and lively imagery that captures attention. |
| **3. Serene Natural Settings** Many images feature serene outdoor environments, including natural landscapes, gardens, and bodies of water, highlighting a calm and peaceful atmosphere. | **3. Vivid and Dynamic Elements** Descriptions focus on colorful and lively scenes, with vibrant attire, festive settings, and active engagements, emphasizing visual appeal and movement. | **3. Close-Up and Detailed Views** Descriptions often emphasize close-up shots, highlighting the intricate details, textures, and designs of objects, with a focus on aesthetic and functional attributes. |
| **4. Detailed Environmental Context** Descriptions typically include detailed elements of the scene, focusing on settings like trees, buildings, and objects in natural or urban environments, emphasizing the rich context and interactions between elements. | **4. Detailed Objects and Clothing** Captions highlight specific objects and items of clothing, such as jewelry and accessories, with intricate details on patterns, colors, and designs. | **4. Serene and Artistic Compositions** Many images showcase static, picturesque environments or artistic designs, creating a calm and detailed visual representation of both natural and structured settings. |
| **5. Emotions and Interactions** Scenes frequently capture moments of celebration, competition, or personal interaction, highlighting the emotions and dynamic aspects of the subjects in various social and communal settings. | **5. Creative and Artistic Themes** There is a notable presence of artistic scenes including paintings, murals, sculptures, and elements of performance and fantasy, showcasing creativity. | **5. Simplistic and Isolated Backgrounds** Scenes are often set against simplistic, plain backgrounds, drawing attention to the main subject and creating a clean, uncluttered look. |

Figure 20: **Full list of LLM summarization** of dataset features.

# B  Additional Results on Dataset Classification with Transformations

## B.1  Handcrafted Features

In Sections 3.2 and 3.3, we primarily leverage modern neural networks to transform images into alternative representations. Here, we apply classic computer vision algorithms to extract handcrafted features and assess whether these features can effectively capture dataset bias.

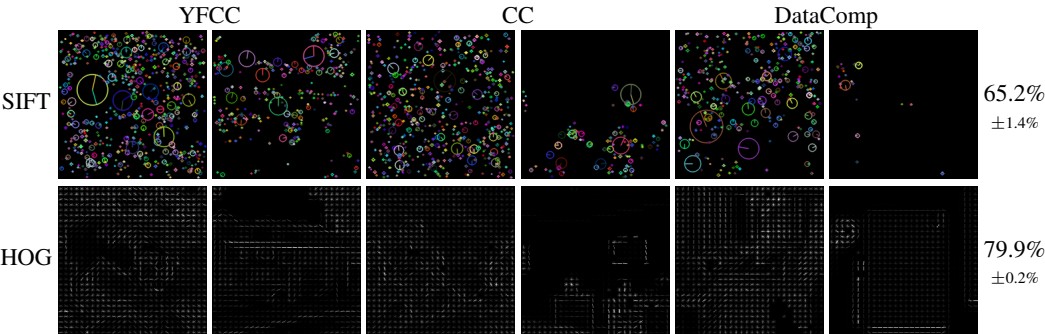

Figure 21: **Transformations encoding handcrafted features**. SIFT keypoints can moderately capture dataset bias, while HOG achieves close-to-reference accuracy by capturing local gradients.

**Scale-Invariant Feature Transform (SIFT)** [43] identifies keypoints that are invariant to scale changes. Each keypoint is characterized by location, scale, and orientation. SIFT computes the Difference of Gaussian (DoG) across multiple scales and identifies keypoints as the local extrema in the DoG images. These keypoints are then refined through a Taylor expansion for precise localization, and orientation is assigned based on local gradient directions. In Figure 21, we visualize each keypoint as a circle, with radius length and direction indicating its size and orientation. The classification accuracy of 65.2% indicates that SIFT features are moderately effective at capturing dataset bias.

**Histograms of Oriented Gradients (HOG)** [14] robustly represents gradient patterns by aggregating gradient information within small spatial regions (cells) into histograms. These histograms are normalized over a block of adjacent cells for better invariance to changes in illumination and contrast. Figure 21 visualizes HOG with the dominant gradient direction in each cell, where brightness indicates gradient magnitude. The near-reference accuracy of 79.9% suggests that the shape and contour features captured by HOG are highly indicative of the dataset identity.

## B.2  Ablation of Different Pre-trained Models

Here we explore how the performance of different pre-trained models used for transformations affects our results in Section 3. Specifically, we perform object detection, contour extraction, and depth estimation using smaller and less powerful models ViTDet-Base, SAM (ViT-Base), and Depth-Anything-V2 (ViT-Base). As shown in Table 3, the pre-trained model size minimally impacts the classification accuracy.

| | ViTDet-B | ViTDet-H | | SAM (ViT-B) | SAM (ViT-L) |
|---|---|---|---|---|---|
| # parameters | 145M | 695M | # parameters | 94M | 312M |
| accuracy | 60.8% | 61.5% | accuracy | 72.3% | 73.1% |

| | Depth-Anything-V2 (ViT-B) | Depth-Anything-V2 (ViT-L) |
|---|---|---|
| # parameters | 98M | 335M |
| accuracy | 72.6% | 73.4% |

Table 3: **Varying pre-trained model size** for object bounding box generation, SAM contour formation, and depth estimation minimally affects dataset classification accuracy on transformed datasets.

Additionally, we apply our LLM summarization method in Section 4.2 to other LLMs: Claude 3.5-Sonnet [2] and Llama-3.1-8B-Instruct [17]. Figures 22 and 23 list the 5 summarized dataset features. The features from different LLMs express highly similar concepts for each dataset.

**1. Outdoor Settings**
Predominantly features natural landscapes, cityscapes, and public spaces, showcasing diverse environments from beaches and forests to urban streets.

**2. Group Dynamics**
Frequently depicts large gatherings and crowded scenes, capturing social interactions and collective activities in various contexts.

**3. Action and Movement**
Often portrays dynamic scenes with people or animals in motion, emphasizing the energy and liveliness of the captured moments.

**4. Wildlife and Nature**
Regularly includes flora and fauna elements, highlighting the presence of animals and natural features within the images.

**5. Visual Details**
Provides rich descriptions of clothing, surroundings, and contextual elements, offering a comprehensive view of the scenes depicted.

**1. Detailed Focus**
Emphasizes specific objects, people, or scenes with technical details and close-up views.

**2. Artistic Representation**
Frequently includes stylized or creative designs, from logos to cartoon characters and abstract compositions.

**3. Clothing and Accessories**
Often provides detailed descriptions of attire, personal items, and their positioning within the image.

**4. Setting Variety**
Covers diverse environments, from intricate indoor layouts to historical scenes and unique scenarios.

**5. Text and Branding**
Commonly features written elements, references to specific brands, characters, or pop culture icons.

**1. Object Focus**
Close-up views of individual products or simple scenes dominate the imagery.

**2. Background**
Items are frequently presented against plain, often white backgrounds for clarity.

**3. Branding**
Product images commonly feature visible brand names, logos, or technical specifications.

**4. Detail Emphasis**
Descriptions highlight specific features, materials, colors, and designs of objects.

**5. Content Variety**
Imagery includes diverse elements like household items, abstract designs, and informational content such as diagrams or book covers.

Figure 22: **Claude 3.5-Sonnet's summarization** of dataset features.

**1. Natural Environments**
The distribution features images of serene and picturesque natural environments, such as landscapes, forests, and mountains.

**2. Everyday Life and Human Connection**
The images depict people engaging in various activities, creating a sense of everyday life and human connection in natural or everyday settings.

**3. Vibrant Colors and Dynamic Compositions**
The images often feature vibrant colors, dynamic compositions, and a sense of energy or movement, capturing the beauty and excitement of the world around us.

**4. Focus on People and Relationships**
The images frequently include people as the main subject, with a focus on their actions, emotions, or interactions with their environment, often conveying a sense of energy, movement, or excitement.

**5. Realistic and Detailed Scenes**
The images often feature everyday scenes, presented in a realistic and detailed manner, showcasing natural environments and ordinary activities.

**1. Detailed Compositions**
The images often feature detailed and composed scenes, with an emphasis on showcasing the textures and details of the subject matter.

**2. Urban Landscapes and Settings**
The images frequently depict urban environments, such as streets, buildings, and cityscapes, capturing the energy and activity of city life.

**3. Stylized and Artistic Visuals**
The images often have a more stylized or artistic quality, with a focus on composition, lighting, and texture, and may include text, logos, or other visual elements that add context or meaning to the scene.

**4. Emphasis on Objects and Concepts**
The images frequently focus on objects, concepts, or ideas, highlighting their significance, functionality, or emotional resonance, often with a mix of objects, people, and text.

**5. Visual Arrangement and Composition**
The images often feature objects, food, or other visual elements arranged in a visually appealing manner, with a focus on color, texture, or composition.

**1. Focus on Color and Texture**
The images often feature a focus on color, shape, or texture, particularly in abstract or conceptual images, and detailed close-up views of objects or visual elements.

**2. Minimalist Backgrounds**
The images frequently have a more minimalist or neutral background, which allows the object or product to be the main focus, often with a focus on showcasing their design, texture, and visual appeal.

**3. Object-Centric Compositions**
The images often feature objects, products, or scenes that are more static or still, such as a product on a white background, a diagram or chart, or a scenic view, with a focus on showcasing their design, texture, and visual appeal.

**4. Attention to Detail**
The images frequently include detailed, close-up views of objects or visual elements, such as intricate designs, patterns, or textures, which suggests a focus on craftsmanship and artistry.

**5. Simple yet Effective Composition**
The images often feature a focus on simple colors, clean backgrounds, and a focus on the main subject or information being conveyed, resulting in a sense of simplicity and minimalism.

Figure 23: **Llama-3.1-8B-Instruct's summarization** of dataset features.

## B.3 Dataset Classification with Different Vision Backbones

In Section 3, we employ ConvNeXt-Tiny as the base classifier to perform the dataset classification task on transformed images. To see how our results change for models with different sizes and architectures, we further train ConvNeXt-Femto [74], ConvNeXt-Nano, and ResNet-34 [26] to perform the dataset classification task for each transformation. In Table 4, the results show that the classification accuracy for each transformation is stable across different models.

| | ConvNeXt-Femto | ConvNeXt-Nano | ResNet-34 | ConvNeXt-Tiny |
|---|---|---|---|---|
| # parameters | 4.8M | 15.0M | 21.3M | 27.8M |
| reference | 79.9% | 81.2% | 81.6% | 81.7% |
| semantic segmentation | 68.8% | 69.8% | 71.7% | 70.3% |
| object detection | 59.8% | 60.1% | 61.7% | 61.5% |
| VAE | 73.7% | 76.2% | 77.3% | 77.1% |
| Canny edge detector | 69.2% | 70.3% | 71.1% | 70.8% |
| SAM contour | 70.6% | 72.3% | 73.8% | 73.1% |
| pixel shuffling (random order) | 52.6% | 52.6% | 60.8% | 52.2% |
| pixel shuffling (fixed order) | 56.6% | 57.7% | 59.5% | 58.5% |
| patch shuffling (random order) | 78.5% | 78.7% | 79.2% | 80.2% |
| patch shuffling (fixed order) | 78.5% | 79.2% | 79.9% | 81.0% |
| RGB mean | 47.9% | 47.9% | 48.0% | 47.9% |
| low-pass filter* | 71.2% | 72.7% | 70.4% | 73.8% |
| high-pass filter* | 76.6% | 78.9% | 81.6% | 79.1% |
| unconditional generation | 77.1% | 79.9% | 81.6% | 76.5% |
| text-conditional generation | 56.8% | 57.3% | 57.9% | 57.8% |

Table 4: **Different image classification models**' validation accuracy on transformed datasets. The accuracy remains consistent across models and transformations. ∗ Note the frequency filters here are Butterworth filters [9] with a threshold of 30.

We also finetune two additional sentence embedding models MPNet-Base [63] and Sentence-BERT-Base [57] for dataset classification on LLaVA-generated captions for YCD images. Note both of these models are weaker [49] than our default Sentence-T5-Base in Section 3.2. Nevertheless, as shown in Table 5, the classification accuracy remains consistent across models and transformations.

| transformation | MPNet-Base | Sentence-BERT-Base | Sentence-T5-Base |
|---|---|---|---|
| short caption | 63.6% | 63.4% | 63.7% |
| long caption | 66.2% | 65.9% | 66.0% |

Table 5: **Different sentence embedding models**' dataset classification accuracy on generated captions of YCD images. The accuracy is still high even on weaker sentence embedding models.

## B.4 Combining Information from Multiple Transformations

We are interested in whether combining several visual attributes can jointly contribute to a larger dataset bias. To this end, we transform each image in the YCD datasets using two different transformations, and concatenate the two resulting images along the channel dimension into a single image. We consider all pairwise combinations of object detection, pixel shuffling, and SAM contour. Table 6 shows the resulting classification accuracies. Combining semantic and structural attributes can result in higher dataset classification accuracy compared to using a single attribute alone.

| transformation 1 (accuracy) | transformation 2 (accuracy) | combined accuracy |
|---|---|---|
| pixel shuffling (52.2%) | object detection (61.5%) | 68.9% |
| pixel shuffling (52.2%) | SAM contour (73.1%) | 74.2% |
| object detection (61.5%) | SAM contour (73.1%) | 73.1% |

Table 6: **Combination of different transformations** can lead to larger bias.

## B.5 Is the Dataset Classification Model Memorizing or Generalizing?

To show that the high validation accuracy on the transformed datasets in Section 3 is achieved through generalization rather than memorization of training examples, we follow Liu and He [40] to perform the dataset classification task on pseudo-datasets. However, unlike Liu and He [40], we construct the pseudo-datasets from transformed images instead of original images.

Specifically, we create three pseudo-datasets, each sampled without replacement from the YFCC dataset applied with one specific transformation. Tables 7 and 8 present the pseudo-dataset classification *training* accuracy on YFCC bounding boxes and YFCC Canny edges, without or with augmentations. As expected, the classification models fail to converge with more training images or stronger augmentations. *All pseudo-dataset classification models have a chance-level validation accuracy of 33%*, as they merely memorize the dataset origin of each training image rather than learning generalizable patterns.

| imgs per set | w/o aug | w/ aug |
|---|---|---|
| 100 | 100% | 100% |
| 1K | 100% | 100% |
| 10K | 100% | fail |
| 100K | fail | fail |

Table 7: Training accuracy for YFCC bounding box pseudo-dataset classification.

| imgs per set | w/o aug | w/ aug |
|---|---|---|
| 100 | 100% | 100% |
| 1K | 100% | 100% |
| 10K | 100% | fail |
| 100K | fail | fail |

Table 8: Training accuracy for YFCC Canny edge pseudo-dataset classification.

## B.6 High-pass and Low-pass Filters at Different Thresholds

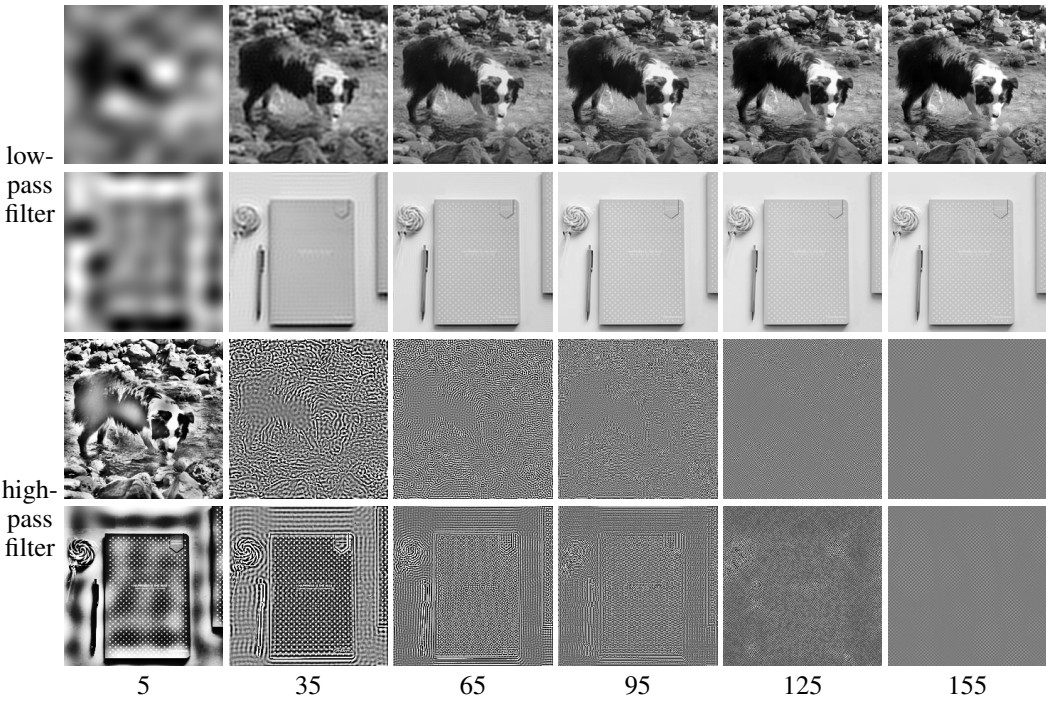

Figure 24: **Ideal filter [22] with different thresholds**. We select filtering thresholds {5, 35, 65, 95, 125, 155}. The high-pass filtered images are equalized for better visualization.

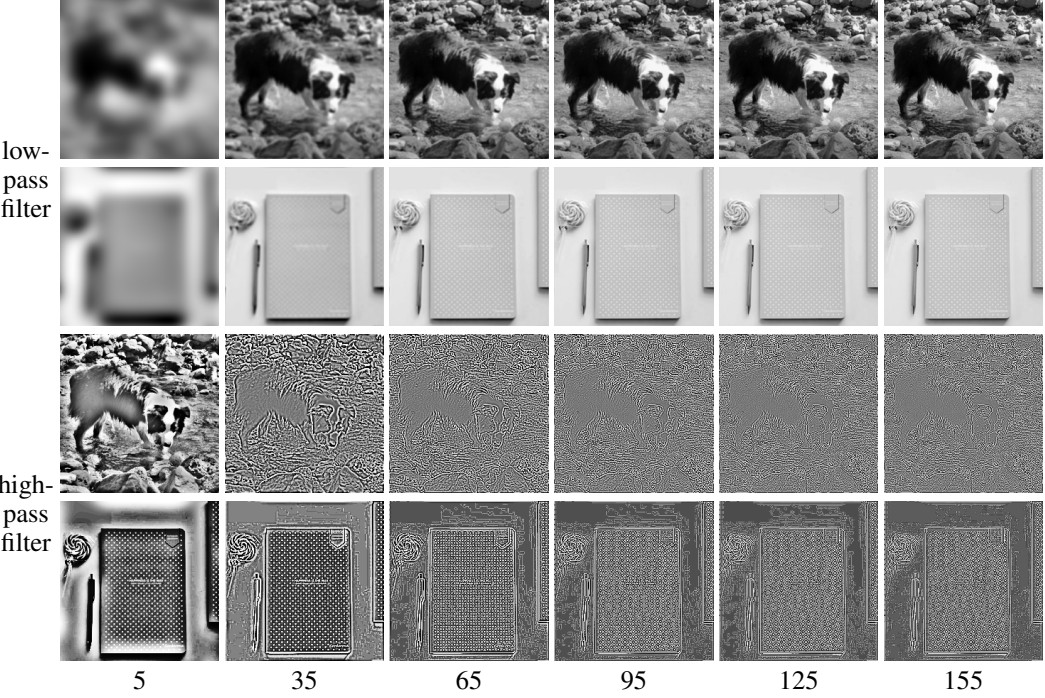

Figure 25: **Butterworth filter [9] with different thresholds**. We select filtering thresholds {5, 35, 65, 95, 125, 155}. The high-pass filtered images are equalized for better visualization.

In Section 3.6, we perform dataset classification on high-pass and low-pass filtered images to quantify the dataset bias in different frequency components. Here, we further investigate how varying thresholds affect classification performance. Figures 24 and 25 display transformed images with ideal filter [22] and Butterworth filter [9]. Figure 26 presents the resulting dataset classification accuracies across different thresholds. While accuracies generally decline at very strict threshold values, in other cases, ideal and Butterworth filters achieve high accuracies for both low-pass and high-pass filters.

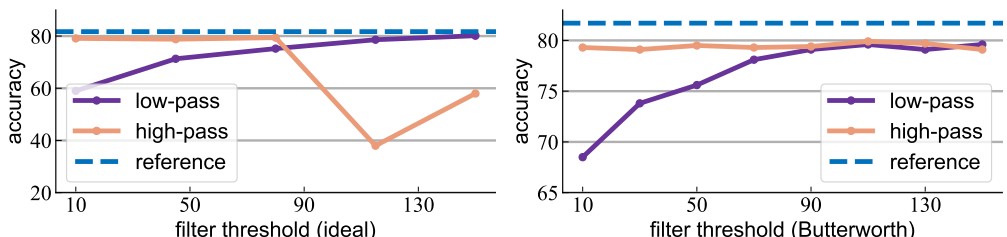

Figure 26: **Accuracy on filtered images at different thresholds**. For most thresholds, both ideal and Butterworth filters achieve high accuracies for both high-pass and low-pass filters.

It is worth noting that the Butterworth filter employs a soft thresholding approach when filtering frequencies, therefore allowing each image to retain some frequency information beyond the selected threshold. This likely contributes to the high accuracy observed on Butterworth high-pass filtered images, even at higher thresholds. Additionally, we observe that the ideal high-pass filter can achieve a higher classification accuracy with a threshold of 150 compared to 115. This is possibly because when the threshold is at 115, the biased higher-frequency information is dominated by less-biased lower-frequency information.

### B.7 Representing Semantic Segmentation Masks with Binary Arrays

In Section 3.2, we apply semantic segmentation to capture semantic information from images and perform dataset classification on the segmentation masks. In particular, we represent the 150-class semantic segmentation mask as an RGB image using a color palette, where each semantic class is assigned a distinct color from the predefined palette. This results in 69.8% accuracy. Alternatively, training the classification model directly on the 150-channel binary segmentation mask yields a slightly lower accuracy of 67.6%. The difference in performance may result from the model processing the compact RGB format more effectively than the sparse, 150-channel binary format.

## C Verifying Semantic Patterns

In Section 4.2, we leverage various language analyses to provide natural language descriptions of each dataset's characteristics. Here, we aim to validate these characteristics with a Vision-Language Model (VLM) and VisDiff [18].

### C.1 Vision-Language Models

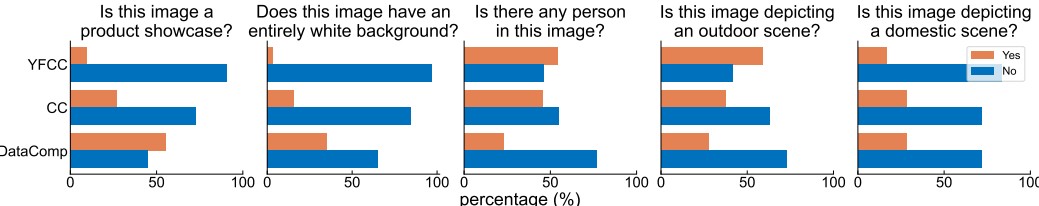

Figure 27: **High-level semantic features' distributions annotated by LLaVA**. The imbalanced distributions across YCD confirm the dataset characteristics in Section 4.2.

To confirm the semantic patterns we found in the YCD datasets, we pick five representative ones and prompt a VLM LLaVA 1.5 [38] to answer whether each pattern exists in images. In Figure 27, we plot the distribution of LLaVa's responses across the YCD datasets. The results quantitatively substantiate the identified dataset themes, further evidencing that (1) DataComp is characterized by product showcase and white backgrounds but lacks human figures and (2) YFCC focuses on outdoor scenes, while CC and DataComp contains more images depicting domestic and indoor environments.

We also manually check the LLaVA annotation quality with a handful of samples. Table 9 displays some examples. The question and images involving human presence are omitted for privacy reasons.

| dataset | YFCC | | CC | | DataComp | |
|---|---|---|---|---|---|---|
| image | | | | | | |
| product showcase | ✓ | ✗ | ✓ | ✗ | ✓ | ✓ |
| white background | ✗ | ✗ | ✓ | ✗ | ✗ | ✓ |
| outdoor scene | ✗ | ✓ | ✗ | ✓ | ✗ | ✗ |
| domestic scene | ✗ | ✗ | ✗ | ✗ | ✓ | ✓ |

Table 9: **Examples of LLaVA annotations**. LLaVA can answer our question with reasonable accuracy. The four rows correspond to the first, second, fourth, and fifth questions in Figure 27.

## C.2 VisDiff

| | YFCC | CC | DataComp |
|---|---|---|---|
| (compared to) YFCC | | • unique home décor
• pictures with elements of food or cooking
• furniture and interiors | • product Images
• images of beauty products
• clothing or fashion items |
| (compared to) CC | • outdoor sports activities
• interactions between people
• outdoor recreational activities | | • furniture and appliances
• Images of sports goods
• objects in black and white photography |
| (compared to) DataComp | • people involved in activities
• interactions between people
• people posing for photos | • people at gatherings
• human focused images
• performing tasks | |

Figure 28: **Dataset features generated by VisDiff [18]**. Each cell lists the top 3 concepts distinguishing the dataset in each column from the one in each row. Note the VisDiff concepts highly overlap with the dataset characteristics from our language analysis in Section 4.2.

VisDiff [18] is a recently proposed method for describing differences between two image sets in natural language. It first uses a VLM to generate image captions, and then leverages an LLM to propose concepts that distinguish the first image set from the second image set based on these captions. For each concept, VisDiff uses the CLIP similarity scores between the images and that concept to classify two image sets, and calculates the AUROC of this classification. The final output concepts are ranked based on their corresponding AUROC scores.

For each dataset pair in the YCD datasets, we use VisDiff to identify the top 5 concepts that better describe the first dataset compared to the second one, as shown in Figure 28. VisDiff's concepts closely align with the dataset characteristics of our language analysis in Section 4.2, highlighting "people" and "outdoor activities" for YFCC and "products" for DataComp. However, our method can also identify specific visual biases across datasets with various transformations, fixed object-level queries, and in-depth natural language analysis.

# D Examples of Long Captions

Table 10 shows examples of long paragraph captions generated by LLaVA 1.5 [38] in Section 3.2.

| dataset | images | long paragraph caption |
|---|---|---|
| YFCC |  | A black and white dog is standing in a shallow body of water, possibly a river or stream. The dog appears to be drinking from the stream, surrounded by rocks. The scene captures the dog's natural behavior as it quenches its thirst at a water source. |
| |  | A food truck is parked on the side of the road, with a man standing nearby. The truck is blue and features a surfboard on the side, indicating a surf theme for its offerings. The man, possibly the owner or an employee, stands beside the truck, either waiting for customers or preparing to serve food. |
| CC |  | A large herd of elephants is walking across a grassy field, with at least 13 elephants visible. Some are closer to the foreground, while others are further in the background, spread out across the field in pairs or individually. The scene captures the majesty of these creatures in their natural habitat. |
| |  | A cozy bedroom featuring a bed, desk, and chair. Above the desk, a TV is mounted on the wall, and a laptop rests on the desk, accompanied by a vase of flowers that adds elegance to the room. The space also includes a dining table and chairs, providing versatility for different activities. |
| DataComp |  | A spacious dining room with a large wooden table surrounded by chairs. The table is set with a white tablecloth, and pillows on the chairs add to the cozy ambiance. Natural light fills the room, and potted plants are placed around the area, enhancing the overall atmosphere. |
| |  | A notebook and pen are placed on a table next to a lollipop. The notebook, open with a polka dot design, lies beside the pen, ready for use. The colorful lollipop adds a playful element to the scene. |

Table 10: **Examples of long paragraph captions** generated from LLaVA 1.5 [38].

# E  Additional Information on ImageNet Object Analysis

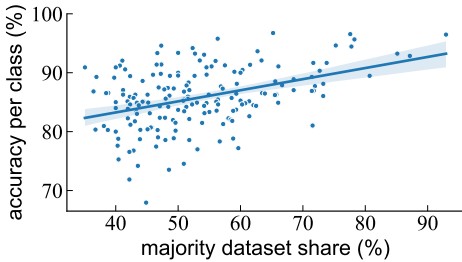

Figure 29: **Majority dataset share for each ImageNet object class** positively correlates with the reference dataset classification accuracy on YCD images within that object class.

The difference in object distribution among datasets based on ImageNet object queries in Section 4.1 can be further used to perform dataset classification. Specifically, we classify an image into the dataset that has the highest frequency of this image's label. Without any learnable parameters (unlike logistic regression in Section 4.1), this simple decision rule achieves a validation accuracy of 50.41%.

The dataset classification model trained on original YCD datasets in Section 3.1 might also leverage the imbalance of underlying object-level distributions. To investigate this, we partition the YCD images by their ImageNet object annotations in Section 4.1. For each object class, we calculate the proportion of its images originating from YFCC, CC, and DataComp, respectively, and define the highest proportion as the *majority dataset share*. Further, we calculate the dataset classification accuracy for images with each ImageNet object annotation. Figure 29 shows that the dataset classification accuracy for each object class is positively correlated with its majority dataset share.

# F  Limitations

Since our framework relies heavily on pre-trained recognition and generative models [58, 12, 36, 34, 39] for extracting semantic and structural information from images, the analysis may be affected by the bias inherent to those models or the datasets they are trained on. For example, if the pre-trained models are trained on data very similar to a certain dataset under study (*e.g.*, one of YCD), the measured level of bias may be affected. In addition, dataset classification can only reveal bias by comparing multiple datasets, and can not be directly applied to a single dataset to understand its bias.

# G  Broader Impacts

Our framework can be used to analyze the datasets before model training, to better determine whether the dataset composition aligns with training goals. It is relatively a fast process compared to a full training cycle. This can help researchers reduce experiment iterations and thus total energy usage.

