# OpenReview forum: "Understanding Bias in Large-Scale Visual Datasets"
_NeurIPS.cc/2024/Conference — NeurIPS 2024 poster_

### Official Review · Reviewer_aLTB · 2024-07-04

**Soundness:** 3
**Presentation:** 3
**Contribution:** 2
**Rating:** 6
**Confidence:** 3

**Summary:**

This paper explores dataset biases and introduces a framework to identify the unique visual attributes that differentiate various datasets. The method involves applying a range of transformations to extract semantic, structural, boundary, color, and frequency information from the datasets, evaluating how each type of information contributes to their distinct characteristics.

**Strengths:**

1. A diverse range of transformations was considered for this study.
2. Provides a comprehensive study to identify various factors that allow the identification of various biases that help to distinguish between datasets.
3. The paper is written well.

**Weaknesses:**

1. Even though the paper identifies various factors that help in distinguishing datasets, it does not provide any information/ takeaways on how this information could be used to 'build more diverse and representative datasets in the future'.I consider the investigative analysis as a good and significant contribution but not commenting on how to utilise these results in a way that would be useful for the community is a significant shortcoming of this paper. The authors could have moved some of the transformations to the appendix and utilise the space for describing 'key takeaways and explaining how &what dataset curators should keep in mind while building datasets.
2. No information is provided on how the accuracy of pre-trained models affects the inferences or findings.
3. There is no discussion on how the format of data transformation affects the findings. For instance, converting the dataset into segmentation masks or object detection boundaries could capture more information in the segmentation results. The authors have not addressed the impact of these different transformation formats on their observed findings.

**Questions:**

1. how does the accuracy of pre-trained models affect the inferences or findings?

**Limitations:**

Yes.

---

> ### Author Rebuttal · Authors · 2024-08-05
>
> We thank the reviewer for the insightful review and the constructive comments. We would like to address your concerns below.
> >w1: Even though the paper identifies various factors that help in distinguishing datasets, it does not provide any information/ takeaways on how this information could be used to 'build more diverse and representative datasets in the future'.
>
> Thank you for your suggestion. We are happy to include more discussion about takeaways for dataset curation.
> Our study provides a general framework for identifying the concrete form of low-level and semantic bias in large-scale datasets. **The identified dataset bias can be used retrospectively to analyze the dataset curation procedure.** For example, on YCD:
> - YFCC contains predominantly outdoor scenes and human interactions.
>
> YFCC samples images solely from Flickr, a platform for user-uploaded photos. As Flickr users primarily share personal photos, landscapes, and social interactions, YFCC images predominantly feature natural scenes and human activities. Moreover, YFCC excludes photos labeled as “screenshots” [1], reinforcing the focus on human-related and natural imagery.
> - DataComp has the lowest number of unique objects per image.
>
> DataComp filters for images with high embedding similarity to ImageNet training examples [2], most of which feature object-centric images. While this empirically leads to higher zero-shot performance of downstream CLIP models, it biases the dataset toward images with lower per-image object diversity.
> - CC and DataComp are significantly brighter and contain more digital graphics and object showcase.
>
> These datasets are collected from the Internet and feature results from search engines [2, 3], which prioritize professionally created content like advertisements, infographics, and digital media. This results in a higher prevalence of digital graphics and brighter images, optimized for visual engagement and online presentation.
>
> >w2: No information is provided on how the accuracy of pre-trained models affects the inferences or findings.
>
> To address your concern, we reran some of the experiments in Sections 3 and 4 using different pre-trained models. **While the accuracy of the pre-trained models slightly affects dataset classification accuracy on the transformed datasets, our results and insights remain unchanged.**
>
> For Section 3, we use (1) a weaker VitDet-Base [4] model to extract bounding boxes, (2) a weaker SAM (ViT-B) [5] model to extract object contours, and (3) a stronger generative model SD-XL [6] to text-conditionally generate images. **Even when using different pre-trained models for transformations, the dataset classification model still achieves high accuracy**, suggesting that semantic differences and object shape variations are important contributors to the bias among YCD.
> ||VitDet-Base (new)|VitDet-Huge|
> |-|-|-|
> |#Parameters|145M|695M|
> |Box AP (LVIS) ↑|43.0%|51.5%|
> |Dataset Classification Acc.|60.8%|61.5%|
>
> ||SAM (ViT-Base) (new)|SAM (ViT-Large)|
> |-|-|-|
> |#Encoder Parameters|91M|308M|
> |Dataset Classification Acc.|65.9%|67.3%|
>
> (Due to resource constraints, the SAM experiments are on 300K training samples.)
> ||SD-2.1|SD-XL (new)|
> |-|-|-|
> |#Parameters|983M|3500M|
> |Generation Performance|low|high|
> |Dataset Classification Acc.|55.1%|57.8%|
>
> For Section 4, we use Claude 3.5-Sonnet [7] and Llama 3.1-8B [8] to derive patterns for each dataset as in Figure 16. **The dataset features summarized from these two LLMs closely resemble those of the original GPT-4o**. Specifically, for YFCC, the emphasis is on "people," "outdoor," and "nature," while for DataComp, the focus is on "white background" and "object focus."
>
> Claude 3.5-Sonnet:
> |YFCC|CC|DataComp|
> |-|-|-|
> |Outdoor Settings|Detailed Focus|Object Focus|
> |Group Dynamics|Artistic Representation|White Background|
> |Action and Movement|Clothing and Accessories|Branding|
> |Wildlife and Nature|Setting Variety|Detail Emphasis|
> |Visual Details|Text and Branding|Content Variety|
>
> Llama 3.1-8B:
> |YFCC|CC|DataComp|
> |-|-|-|
> |Natural Environments|Detailed Compositions|Focus on Color and Texture|
> |Everyday Life and Human Connection|Urban Landscapes and Settings|Minimalist Backgrounds|
> |Vibrant Colors and Dynamic Compositions|Stylized and Artistic Visuals|Object-Centric Compositions|
> |Focus on People and Relationships|Emphasis on Objects and Concepts|Attention to Detail|
> |Realistic and Detailed Scenes|Visual Arrangement and Composition|Simple yet Effective Composition|
>
> We will conduct more experiments with other segmentation models and object detection models and add the results and discussion to our draft.
> >w3: There is no discussion on how the format of data transformation affects the findings.
>
> Thank you for pointing this out. We acknowledge that the impact of the transformed images’ format could benefit from further elaboration and a more focused discussion in Section 3:
>
> - Semantic segmentation and object detection: Semantic segmentation provides fine-grained per-pixel semantic annotation, whereas object detection only captures coarse-grained spatial information through bounding boxes. The lack of detailed spatial information contributes to the lower dataset classification accuracy on bounding boxes (61.5%) compared to segmentation masks (67.5%).
>
> - Caption: Caption discards all spatial information, creating more discriminative semantic representations through natural language. This textual representation is less affected by the low-level and spatial cues in the images.
>
> - Edge detection and SAM contour: Object contour delineates fine-grained object shape and spatial information, while lacking the rich object semantic information present in semantic segmentation masks and bounding boxes.

---

> > ### Comment · Reviewer_aLTB · 2024-08-10
> > **Reply**
> >
> > Thanks for the feedback.
> >
> > I think the authors misunderstood my initial review, I didn't mean how "the identified dataset bias can be used retrospectively to analyze the dataset curation procedure". I asked **how these findings can be used to  'build more diverse and representative datasets in the future'?** .

---

> > > ### Author Response · Authors · 2024-08-12
> > > **Reply to "build more diverse and representative datasets in the future"**
> > >
> > > Thank you for your clarifications. Indeed, our previous response focused more on past efforts. **Here we provide several ways our framework can be used to help build more diverse and representative datasets in the future:**
> > >
> > > - When considering adding a new set of images (e.g., from another website) into a data collection, we can first treat new images as a separate dataset, use the transformation and classification framework to tell where and how much they differ from the existing image collections. This can help us decide whether to join them. If the goal is to enhance diversity of the dataset in a certain aspect (e.g., object types), we should only join them when they are sufficiently different.
> > >
> > > - Our language-based analysis provides textual descriptions for any new dataset. This text description directly gives the data curators the intuition on the gist of the dataset, especially when compared with reference datasets. It can help curators refine text terms for image search and tag filtering, in search engines or other platforms.
> > >
> > > - Our framework can identify bias and distribution imbalance in the image statistics (e.g., colors and object distribution). This can help guide adding/removing images with desired/undesired statistics for more balance.
> > >
> > > - The result of a dataset classifier trained on the transformed datasets can serve as a measure of the image's "typicality" within the dataset. For example, images in dataset A that are misclassified as images in dataset B are considered "not typical" within A, for that transformation/attribute. If needed, images with less typical attributes can be then oversampled in collecting data and/or training to enhance representation.
> > >
> > > **In addition to how to use the framework, we list a few direct lessons we learned from our analysis on YCD, that can also help build more diverse and representative datasets in the future:**
> > >
> > > - Filtering by embedding similarity to images of a reference dataset could inherit bias of that dataset.
> > >
> > > We observe that DataComp has the lowest number of unique objects per image (Fig 12). This potentially resulted from DataComp filtering for images with high embedding similarity to ImageNet training examples, most of which feature object-centric images [1].
> > >
> > > To mitigate this, dataset curators should be mindful of the inherent biases in their reference datasets (e.g., ImageNet). Concretely, to enhance the object diversity within DataComp, one could consider using a reference dataset with a higher per-image object diversity (e.g., COCO) during filtering.
> > >
> > > - The source website's image collection mechanism can introduce bias.
> > >
> > > We also noted that YFCC is heavily skewed towards outdoor scenes and human interactions (Sec 4.2). This bias likely stems from its reliance on a single data source, Flickr, where user-generated content often focuses on personal photos, landscapes, and social interactions. Dataset curators should recognize that the collection methods (e.g., user uploads) of data sources (e.g., Flickr) can introduce biases into the resulting dataset (e.g., YFCC).
> > >
> > > - Web-scraped images would naturally contain more digital graphics.
> > >
> > > Since CC and DataComp are crawled from the Internet, they feature results from search engines [2, 3]. This prioritizes professionally created content like advertisements, infographics, and digital media. Curators should evaluate whether this composition aligns with their downstream goals.
> > >
> > > Thank you for your quick reply/clarification. We will add a discussion on this to the paper and see it as an important improvement. We also hear your suggestion on moving experiments to the appendix for the space. Given that NeurIPS policy allows an additional page for accepted publications, we would be able to add the discussion while maintaining the current experiments. We are happy to address any further concerns.
> > >
> > > References:\
> > > [1] Barbu et al, ObjectNet: A large-scale bias-controlled dataset for pushing the limits of object recognition models\
> > > [2] Changpinyo et al, Conceptual 12M: Pushing Web-Scale Image-Text Pre-Training To Recognize Long-Tail Visual Concepts\
> > > [3] Gadre et al, DataComp: In search of the next generation of multimodal datasets

---

> > > > ### Author Response · Authors · 2024-08-13
> > > > **Invitation to discussion**
> > > >
> > > > Dear Reviewer aLTB,
> > > >
> > > > We would like to invite you for any further discussion, as the reviewer-author discussion stage is approaching the end (in 1 day). We hope our response can address your previous concern on how "our findings can be used to build more diverse and representative datasets in the future". We want to make sure we have the chance to clarify/respond again if you have any further concerns. Thank you!
> > > >
> > > > Best,\
> > > > Authors

---

> ### Author Response · Authors · 2024-08-07
> **Rebuttal**
>
> We thank you again for your valuable feedback and we hope our response can address your questions. If you have any further questions or concerns, we are very happy to answer.
>
> [1] Thomee et al, YFCC100M: The New Data in Multimedia Research\
> [2] Gadre et al, DataComp: In search of the next generation of multimodal datasets\
> [3] Changpinyo et al, Conceptual 12M: Pushing Web-Scale Image-Text Pre-Training To Recognize Long-Tail Visual Concepts\
> [4] Li et al, Exploring Plain Vision Transformer Backbones for Object Detection\
> [5] Kirillov et al, Segment Anything\
> [6] Podell et al, SDXL: Improving Latent Diffusion Models for High-Resolution Image Synthesis\
> [7] Anthropic, Claude 3.5-Sonnet\
> [8] Meta, Llama 3.1

---

> ### Comment · Reviewer_aLTB · 2024-08-13
> **Reply**
>
> My concerns are mostly addressed, it would be interesting to see how these suggestions could be incorporated to build a new dataset curation method in future work. I increase my score to 6 after reading the rebuttal.

---

> ### Author Response · Authors · 2024-08-14
> **Thank you**
>
> Many thanks for your valuable feedback and discussion!

---

### Official Review · Reviewer_n4tv · 2024-07-10

**Soundness:** 3
**Presentation:** 3
**Contribution:** 3
**Rating:** 6
**Confidence:** 4

**Summary:**

This work theorizes and investigates various concrete forms of inter-dataset biases, namely those among YFCC, CC, and DataComp (YCD). The authors analyze such inter-dataset biases in pure visual attributes as well as in semantic attributes using LLM-generated descriptions.

**Strengths:**

The paper is a fluent read composed of a clear progressive structure. The authors have shown meticulous effort in constructing the investigations as comprehensive as possible.

**Weaknesses:**

This work extends the scope of [1] and aims to provide specific insights for constructing less biased data collection in future. However, I feel there are several missing evidence that undermine the contributions of this work in its current state:

1. **The authors have only examined the factors of biases individually.** However, I notice that none of the single attributes listed contributes to a higher prediction accuracy over the baseline of using original visual features. Could it be possible that a combination of multiple visual/semantic attributes jointly contributes to the large overall bias? This needs to be verified.

2. **The authors have only investigated one classification model.** According to [1], the inter-dataset overall bias (high prediction accuracy) is observed with multiple classification models as well. So if we use the smaller ResNet-50 model, can we still observe the high accuracy over each individual structural or semantic attribute? I haven't seen the authors ruling out the confounding factor of the classification model size.

3. **The proposed framework consists of indicative bias metrics only relative to YCD.** However, YCD all have their own intra-dataset biases. So when we are attempting to create a debiased dataset in future, *how can we make sure our newly collected data are truly diversified and fair?* Can the authors provide similar evidence over 'Memorization vs. Generalization' as in [1] Sec. 4.2., and verify individual inter-dataset bias attributes with a truly unbiased pseudo-dataset?

[1] A Decade’s Battle on Dataset Bias: Are We There Yet? Liu et al. 2024.

**Questions:**

Please find my major questions in the Weakness section.

Another slight concern is about the presentation. Since an ideal debiased dataset should enjoy an equal likelihood to be predicted as one of the anchor datasets (e.g. YCD), maybe the authors should explicitly clarify this important task setting as early as possible in the paper.

(Aug 9th): Updating my overall ratings thanks to the quality response.

**Limitations:**

The limitations have been sufficiently addressed.

---

> ### Author Rebuttal · Authors · 2024-08-07
>
> We thank the reviewer for the review and the insightful questions. We would like to address your concerns below.
>
> >w1: Could it be possible that a combination of multiple visual/semantic attributes jointly contributes to the large overall bias? This needs to be verified.
>
> Thank you for this suggestion. To verify this, we consider different combinations of 3 transformations with lower dataset classification accuracies: object detection (61.5%), random-order pixel shuffle (52.2%), and SAM contour (73.1%). For each pair of transformations, we combine them by concatenating the resulting transformed images on the channel dimension.
> ||Transformation 1 Acc.|Transformation 2 Acc.|Combined Acc.|
> |-|-|-|-|
> |Obj. Det. + Pixel shuffle|61.5%|52.2%|68.9%|
> |Pixel Shuffle + SAM contour|52.2%|73.1%|74.2%|
> |Obj. Det. + SAM contour|61.5%|73.1%|73.1%|
>
> We observed that **combining semantic and structural attributes resulted in higher dataset classification accuracy compared to using a single attribute alone.**
> >w2: So if we use the smaller ResNet-50 model, can we still observe the high accuracy over each individual structural or semantic attribute?
>
> We are happy to provide more results on this. We’ve expanded our main experiments to employ smaller vision architectures: ResNet-50 (23M parameters) [1] and ConvNeXT-femto (5M parameters) [2]:
> ||baseline|Obj. Det.|SAM Contour|Patch Shuf.|High-pass|
> |-|-|-|-|-|-|
> |ConvNeXt-Tiny (28M)|81.7%|61.5%|73.1%|80.3%|79.4%|
> |ResNet-50 (25M)|82.0%|61.6%|73.6%|80.2%|81.9%|
> |ConvNeXt-Femto (5M)|79.9%|59.8%|70.8%|78.7%|76.7%|
>
> Additionally, we use two weaker sentence embedding models (MPNet-Base [3] and Sentence-BERT-Base [4]) for caption classification:
> ||Short Cap.|Long Cap.|
> |-|-|-|
> |Sentence-T5|63.7%|66.0%|
> |MPNet|63.2%|64.5%|
> |Sentence-BERT|63.2%|65.3%|
>
> **Across model sizes and architectures, the classification accuracy over each individual structural or semantic attribute remains high.** We’ve added this result to our Appendix.
>
> >w3: So when we are attempting to create a debiased dataset in future, how can we make sure our newly collected data are truly diversified and fair? Can the authors provide similar evidence over 'Memorization vs. Generalization' as in [1] Sec. 4.2., and verify individual inter-dataset bias attributes with a truly unbiased pseudo-dataset?
>
> We agree that our framework can only identify bias relative to the given combination of datasets (e.g., YCD) rather than their bias to the universal vision world. **Nevertheless, identifying inter-dataset bias can still be helpful in creating a debiased dataset, which is 'truly diversified and fair', in the future.** For example, in the paper, we used LLM to summarize specific characteristics of each dataset. These characteristics can be used retrospectively to analyze the dataset curation procedure. On YCD:
>
> - YFCC contains predominantly outdoor scenes and human interactions.
>
> YFCC samples images solely from Flickr, a platform for user-uploaded photos. As Flickr users primarily share personal photos, landscapes, and social interactions, YFCC images predominantly feature natural scenes and human activities. Moreover, YFCC excludes photos labeled as “screenshots” [5], reinforcing the focus on human-related and natural imagery.
>
> - DataComp has the lowest number of unique objects per image.
>
> DataComp filters for images with high embedding similarity to ImageNet training examples [6], most of which feature object-centric images. While this empirically leads to higher zero-shot performance of downstream CLIP models, it biases the dataset toward images with lower per-image object diversity.
>
> - CC and DataComp are significantly brighter and contain more digital graphics and object showcase.
>
> These datasets are collected from the Internet and feature results from search engines [6, 7], which prioritize professionally created content like advertisements, infographics, and digital media. This results in a higher prevalence of digital graphics and brighter images, optimized for visual engagement and online presentation.
>
>
> **We observe similar trends over 'Memorization vs. Generalization' as in Sec. 4.2 of [8] on our transformed datasets.** Specifically, we create 3 pseudo-datasets, all of which are sampled without replacement from the same transformed YFCC dataset (e.g., images of Canny-detected edges or images of object bounding boxes). We perform dataset classification on these pseudo-datasets. The tables below present the pseudo-dataset classification _training_ accuracy on YFCC bounding boxes and YFCC Canny-edges. As the number of training images from each dataset increases, the task becomes harder. _All of the models trained on this pseudo-dataset classification have a chance-level accuracy of 33% in the validation set_. This is because they merely memorize the dataset origin of each training image rather than learning any generalizable patterns.
>
> |#Training Images per YFCC Bounding Box Pseudo-Dataset|without augmentation|with augmentation|
> |-|-|-|
> |100|100%|100%|
> |1K|100%|100%|
> |10K|100%|fail|
> |100K|fail|fail|
>
> |#Training Images per YFCC Canny edge Pseudo-Dataset|without augmentation|with augmentation|
> |-|-|-|
> |100|100%|100%|
> |1K|100%|100%|
> |10K|100%|fail|
> |100K|fail|fail|
>
>
> >q1: Another slight concern is about the presentation. Since an ideal debiased dataset should enjoy an equal likelihood to be predicted as one of the anchor datasets (e.g. YCD), maybe the authors should explicitly clarify this important task setting as early as possible in the paper.
>
> We appreciate your suggestion. We’ve added a clarification that “An ideal unbiased dataset should have a chance-level probability of being predicted as any of the anchor datasets.” in our introduction.

---

> ### Author Response · Authors · 2024-08-07
> **Rebuttal**
>
> We thank you again for your valuable feedback and we hope our response can address your questions. If you have any further questions or concerns, we are very happy to answer.
>
> [1] He et al, Deep Residual Learning for Image Recognition\
> [2] Liu et al, A ConvNet for the 2020s\
> [3] Song et al, MPNet: Masked and Permuted Pre-training for Language Understanding\
> [4] Reimers & Gurevych, Sentence-BERT: Sentence Embeddings using Siamese BERT-Networks\
> [5] Thomee et al, YFCC100M: The New Data in Multimedia Research\
> [6] Gadre et al, DataComp: In search of the next generation of multimodal datasets\
> [7] Changpinyo et al, Conceptual 12M: Pushing Web-Scale Image-Text Pre-Training To Recognize Long-Tail Visual Concepts\
> [8] Liu et al, A Decade’s Battle on Dataset Bias: Are We There Yet?

---

> ### Comment · Reviewer_n4tv · 2024-08-10
> **This is some impressive response.**
>
> I greatly appreciate the authors effort in putting up the response. In fact, it's beyond my expectation. I find my 3 major concerns have been adequately addressed thanks to the additional information - I trust the authors will incorporate them into the final version.
>
>
> In retrospect, I believe what hindered my initial impression was the title of the paper, which led me to believe this was some incremental work over investigating *'intra-dataset biases'*, such as imbalanced distribution of object classes or visual features and etc. I highly recommend, if applicable, that the authors explicitly open up with **'INTER-dataset biases'** in the title or somewhere early-on in the revised paper. Overall, I am willing to update my rating owning to this thorough rebuttal.

---

> ### Author Response · Authors · 2024-08-10
> **Thank you**
>
> Thank you again for your helpful comments and for reviewing our response! We are glad to hear that the concerns have been addressed. We will incorporate the additional results discussed in our rebuttal into the next version of the paper and emphasize that we're studying the inter-dataset bias in our introduction to better reflect the focus of our work.

---

### Official Review · Reviewer_yVzd · 2024-07-13

**Soundness:** 4
**Presentation:** 4
**Contribution:** 3
**Rating:** 8
**Confidence:** 4

**Summary:**

This paper studies the problem of dataset bias prevalent in current multimodal datasets. It revisits the dataset classification experiment from Torralba et al, recently studied again by Liu and He, and deconstructs their findings to understand what aspects of datasets (structural, semantic, color, object-level, caption-level etc) contribute most to the bias prevalent in visual datasets. With abundant empirical evidence, the paper provides an interesting and important analysis on the fundamental gaps in our understanding of large-scale image-text datasets.

**Strengths:**

- The paper is very well written and presented, the research question is concisely stated, and all the experimental results clearly tie back with the main research question.
- The research question studied in itself is on that is pivotal in the current age of foundation models. Understanding datasets is key for understand models, and this paper takes a few steps to further our understanding of these giant datasets.

**Weaknesses:**

I have a few technical concerns that might undermine the significance of this paper's findings, I note them down here. Further, I also have a few suggested additional experiments that might help boost the significance of this paper's results.

- All the results rely on a three class classification problem at heart. How should one interpret these results under the argument that neural networks in general can learn noisy labels, and can fit arbitrary distributions [1]? How does the baseline 81.7% performance change if you assigned random labels to the original set of 3M images? This is a very important ablation to verify the significance of all the findings.

- One issue with the semantic segmentation classification experiment (sec 3.2) is that it removes the pixel information from the task, and converts it into a potentially much simpler task where the input features are only of size 150 (if I understand the experiment correctly). Here is a simple suggestion for removing that confounder. What happens if you train directly on the semantic masks itself? Of-course there would be some bias coming in from the colour applied to the segmentation masks for each of the different object classes, but this could be mitigated by training two different models using two different mask colour palettes, and looking at the variance. If the accuracy of this particular task still remains similarly high, that would further boost the significance of the results.

- For the caption classification task (sec 3.2), could you also use another sentence embedding model that is potentially weaker in its initial MTEB [2] performance, for reproducing the results of that experiment. Again, this would help boost the significance of the findings.

- Would the main results be consistent across two completely different subsets of the YCD classificaiton task? Could you sample an entirely different set of 3M images and 30K validation images, rerun the experiments, and cross-validate on the two different validation sets?

- For figure 11, could you redo that analysis on another subset of 3M samples as well? I am skeptical that this result would hold exactly true for a given dataset, especially given this prior result that most datasets curated from the web have similar concept distributions [3]. Can the authors provide a reconciliation between this prior result and their results?

- Another interesting analysis that can be done would be to use the VisDiff tool [4] to perform an analysis on the visual distribution differences between the pairs of YCD datasets. It might provide an interesting sub-analysis and would help further verify the results in sec 4.2 and fig. 16 in the paper.

- The paper has a lot of very interesting insights about bias across visual datasets. However, there are no concrete suggestions/discussion on how one could go about mitigating these biases. A discussion regarding the source of these datasets, different data curation / filtering mechanisms, and how they might potentially impact these biases, would make for a great and required addition to this paper.

[1] Zhang et al, Understanding deep learning requires rethinking generalization
[2] Muenighoff et al, MTEB: Massive Text Embedding Benchmark
[3] Udandarao et al, No "Zero-Shot" Without Exponential Data: Pretraining Concept Frequency Determines Multimodal Model Performance
[4] Dunlap et al, Describing Differences in Image Sets with Natural Language

**Questions:**

All my questions are also mentioned in the weaknesses section above.

**Limitations:**

Yes the authors have mentioned that their main weakness is with respect to using pretrained models which themselves might be biased.

---

> ### Author Rebuttal · Authors · 2024-08-05
>
> We sincerely thank you for your constructive comment. We are encouraged that you find our paper provides helpful insights about bias in large-scale datasets. We address your concerns below:
> >w1: How should one interpret these results under the argument that neural networks in general can learn noisy labels, and can fit arbitrary distributions? How does the baseline 81.7% performance change if you assign random labels to the original set of 3M images?
>
> All the reported accuracies in our work are evaluated on 30k validation samples. On the other hand, **the findings in [1] about high model accuracy on random labels through memorization are only on the _training_ set.** When we assign random labels to our data, the model only achieves a chance-level accuracy of 33.3% on the validation set. This is expected since models trained on random labels cannot learn dataset-specific patterns that is generalizable to validation sets.
> >w2: What happens if you train directly on the semantic masks itself?
>
> **We would like to clarify that we converted each pixel, rather than the full image, to a 150-channel binary array** (Line 79). The transformed image maintains the original width and height.
>
> Additionally, training on the _RGB_ segmentation masks with 2 different color palettes results in stable accuracies of 70.1% and 70.0%.
> >w3: For the caption classification task (sec 3.2), could you also use another sentence embedding model that is potentially weaker in its initial MTEB performance?
>
> We used weaker models MPNet-Base [2] and Sentence-BERT-Base [3] for caption classification:
> ||MPNet|Sentence-BERT|Sentence-T5|
> |-|-|-|-|
> |Short Cap.|63.2%|63.2%|63.7%|
> |Long Cap.|64.5%|65.3%|66.0%|
>
> **The accuracy is consistent on all 3 sentence embedding models.**
> >w4: Could you sample an entirely different set of 3M images and 30K validation images, rerun the experiments, and cross-validate on the two different validation sets?
>
> We reran the experiments on new samples (3M training, 30K validation). Due to time constraints, caption and segmentation experiments are on 300k new training samples. **Accuracies vary minimally across two samples.**
> ||baseline|Short Cap. (300k)|Long Cap. (300k)|Sem. Seg. (300k)|Obj. Det.|Patch Shuf.|Canny|High-pass|
> |-|-|-|-|-|-|-|-|-|
> |train (ori.)>val (ori.)|81.7%|61.5%|63.1%|54.6%|61.5%|80.3%|70.84%|79.4%|
> |train (ori.)>val (new)|82.0%|61.8%|63.5%|55.3%|62.2%|80.3%|70.8%|79.3%|
> |train (new)>val (ori.)|82.1%|61.3%|63.1%|54.6%|61.0%|79.8%|70.4%|80.1%|
> |train (new)>val (new)|82.1%|61.6%|63.6%|55.1%|62.2%|79.9%|71.12%|80.3%|
>
> >w5: For Fig 11, could you redo that analysis on another subset of 3M samples as well? … Can the authors provide a reconciliation between this prior result and their results?
>
> We observe **the number of overlapped top 10 object categories (as in Fig 11) is high between the original 3M and new 3M images**:
> |#Overlaps|YFCC|CC|DataComp|
> |-|-|-|-|
> |ImageNet|9|8|8|
> |LVIS|10|10|10|
> |ADE20k|10|9|9|
>
> This does not contradict [4]. Fig 11 shows the top 10 object classes with the highest proportion of their images from a particular dataset, rather than the most frequent object classes within each dataset. Despite certain object categories being overrepresented in certain datasets, the overall object distribution vectors [4] are highly correlated:
>
> |ADE20k Corr |YFCC|CC|DataComp|
> |-|-|-|-|
> |YFCC|1|0.92 |0.82|
> |CC||1|0.97|
> |DataComp|||1|
>
> |LVIS Corr|YFCC|CC|DataComp|
> |-|-|-|-|
> |YFCC|1|0.90|0.81|
> |CC||1|0.93|
> |DataComp|||1|
> >w6: Another interesting analysis ... would be to use the VisDiff tool to perform an analysis on the visual distribution differences between the pairs of YCD datasets.
>
> Thank you for recommending VisDiff [5]. We generated caption-based set differences for each dataset pair (**Figure 2 in the attached pdf**):
> ||Y|C|D|
> |-|-|-|-|
> |(compared to) Y||unique home decor|Product Images|
> |(compared to) C|outdoor sports activities||furniture and appliances|
> |(compared to) D|People involved in activities|people at gatherings||
>
> **Results from VisDiff highly overlap with LDA- and LLM-extracted dataset characteristics**, emphasizing “people” and “outdoor activities” for YFCC and “product” for DataComp. We’ve added this to our Appendix.
> >w7: There are no concrete suggestions/discussion on how one could go about mitigating these biases.
>
> Our study provides a general framework for identifying the concrete form of low-level and semantic bias in large-scale datasets. **The identified dataset bias can be used retrospectively to analyze the dataset curation procedure.** For example, on YCD:
> - YFCC contains predominantly outdoor scenes and human interactions.
>
> YFCC samples images solely from Flickr, a platform for user-uploaded photos. As Flickr users primarily share personal photos, landscapes, and social interactions, YFCC images predominantly feature natural scenes and human activities. Moreover, YFCC excludes photos labeled as “screenshots” [6], reinforcing the focus on human-related and natural imagery.
> - DataComp has the lowest number of unique objects per image.
>
> DataComp filters for images with high embedding similarity to ImageNet training examples [7], most of which feature object-centric images. This biases the dataset toward images with lower per-image object diversity.
> - CC and DataComp are significantly brighter and contain more digital graphics and object showcase.
>
> These datasets are collected from the Internet and feature results from search engines, which prioritize professionally created content like advertisements, infographics, and digital media. This results in a higher prevalence of digital graphics and brighter images, optimized for visual engagement and online presentation.
>
> We have consolidated the above as a discussion paragraph in our draft.

---

> ### Author Response · Authors · 2024-08-07
> **Rebuttal**
>
> We thank you again for your valuable feedback and we hope our response can address your questions. If you have any further questions or concerns, we are very happy to answer.
>
> [1] Zhang et al, Understanding deep learning requires rethinking generalization\
> [2] Song et al, MPNet: Masked and Permuted Pre-training for Language Understanding\
> [3] Reimers & Gurevych, Sentence-BERT: Sentence Embeddings using Siamese BERT-Networks\
> [4] Udandarao et al, No "Zero-Shot" Without Exponential Data: Pretraining Concept Frequency Determines Multimodal Model Performance  \
> [5] Dunlap et al, Describing Differences in Image Sets with Natural Language\
> [6] Thomee et al, YFCC100M: The New Data in Multimedia Research\
> [7] Gadre et al, DataComp: In search of the next generation of multimodal datasets\
> [8] Changpinyo et al, Conceptual 12M: Pushing Web-Scale Image-Text Pre-Training To Recognize Long-Tail Visual Concepts

---

> ### Comment · Reviewer_yVzd · 2024-08-10
> **Response to rebuttal**
>
> I thank the authors for their added experimental results and comments.
>
> (1) The random labels experiment sufficiently answers my question regarding the specificity of these dataset-level biases.
>
> (2) Thanks for running the sentence embeddings and segmentation mask experiments, those high stable accuracies clarify my concern.
>
> (3) The cross-validation results are very strong. I would encourage the authors to somehow incorporate these results into the main paper perhaps as a mean +/- std or something similar, I think this would significantly increase the confidence in the paper's findings.
>
> (4) Thanks for the clarifying discussion b/w your work and the No Zero-shot paper. You being able to reproduce the findings of that paper are very interesting.
>
> (5) Great that the VisDiff results corroborate your analysis.
>
> (6) Your response to w7 is very insightful, thank you for adding this. I would encourage to expand on this discussion and if possible add some sample images from each of the datasets to that section in the appendix. Would be super useful!
>
> Overall, the author rebuttal has significantly increased my confidence in the paper and the main results and insights provided in the paper are very useful. I am updating my score to an 8.

---

> ### Author Response · Authors · 2024-08-12
> **Thank you**
>
> Thank you for your time and feedback on the paper! We’re glad that your concerns have been resolved. We will make sure to include the additional results from the rebuttal in the next version of our paper. The review has been very helpful in enhancing our paper.

---

### Author Rebuttal · Authors · 2024-08-07

Dear Reviewers:

Thanks for all your constructive comments! We hope our response and additional results can address your concerns. Please let us know if you have further questions or comments and we would be more than happy to discuss.

For reviewer yVzd, please note the PDF file attached contains figures in response to weakness 5 and weakness 6.

Best,\
Authors

---

### Decision · Program_Chairs · 2024-09-25

**Decision:**

Accept (poster)

**Comment:**

All reviewers were positive about the paper after the rebuttal. The authors have done substantial new work (experiments, analyses, etc.) in the rebuttal stage, which helped convince the reviewers. AC suggests the authors incorporate them into the revised paper.